# Theories and Methods for Indoor Positioning Systems: A Comparative Analysis, Challenges, and Prospective Measures

**DOI:** 10.3390/s24216876

**Published:** 2024-10-26

**Authors:** Tesfay Gidey Hailu, Xiansheng Guo, Haonan Si, Lin Li, Yukun Zhang

**Affiliations:** 1Department of Software Engineering, Addis Ababa Science and Technology University, Addis Ababa 16417, Ethiopia; tesfaygidey21@std.uestc.edu.cn; 2Department of Information and Communication Engineering, University of Electronic Science and Technology of China, Chengdu 611731, China; sihaonan@std.uestc.edu.cn (H.S.); 201811011926@std.uestc.edu.cn (L.L.); yukunzhang@std.uestc.edu.cn (Y.Z.)

**Keywords:** indoor positioning, comparative analysis, IoT, data fusion, transfer learning, feature engineering, ensemble learning, system design

## Abstract

In the era of the Internet of Things (IoT), the demand for accurate positioning services has become increasingly critical, as location-based services (LBSs) depend on users’ location data to deliver contextual functionalities. While the Global Positioning System (GPS) is widely regarded as the standard for outdoor localization due to its reliability and comprehensive coverage, its effectiveness in indoor positioning systems (IPSs) is limited by the inherent complexity of indoor environments. This paper examines the various measurement techniques and technological solutions that address the unique challenges posed by indoor environments. We specifically focus on three key aspects: (i) a comparative analysis of the different wireless technologies proposed for IPSs based on various methodologies, (ii) the challenges of IPSs, and (iii) forward-looking strategies for future research. In particular, we provide an in-depth evaluation of current IPSs, assessing them through multidimensional matrices that capture diverse architectural and design considerations, as well as evaluation metrics established in the literature. We further examine the challenges that impede the widespread deployment of IPSs and highlight the potential risk that these systems may not be recognized with a single, universally accepted standard method, unlike GPS for outdoor localization, which serves as the golden standard for positioning. Moreover, we outline several promising approaches that could address the existing challenges of IPSs. These include the application of transfer learning, feature engineering, data fusion, multisensory technologies, hybrid techniques, and ensemble learning methods, all of which hold the potential to significantly enhance the accuracy and reliability of IPSs. By leveraging these advanced methodologies, we aim to improve the overall performance of IPSs, thus paving the way for more robust and dependable LBSs in indoor environments.

## 1. Introduction

The rapid proliferation of smartphones and the ubiquity of wireless communication networks in recent years, as well as the emergence of both the fifth- and sixth-generation (5G and 6G) communication systems and the IoT, have fostered a wide range of services, such as localization, tracking, and navigation systems. Due to the incredible development of mobile applications, LBSs have gained significant importance in both industrial and commercial applications. These include indoor navigation [1,2], self-driving cars [3], complex venue management [4], military use [5], commercial location services, hospitals, emergency services, indoor parking [6,7,8], multimedia geotagging, tracking and tourism, forest firefighter tracking and location [9], and many other areas where mobility is important [10,11,12,13]. Global navigation satellite systems (GNSS), such as the GPS, are considered the standard method for positioning and are known for their reliable and promising accuracy in an outdoor environment. However, their applications are limited to indoor positioning because the complex nature of the indoor environment is mainly characterized by (i) unavailability of direct line-of-sight (LOS) or non-line-of-sight (NLOS), resulting in incoherent propagation, caused by various barriers along the transmitting and receiving equipment, (ii) dynamic environments leading to signal variations in both time and space, (iii) severe signal attenuation or shielding of satellite signals [14,15,16,17], and (iv) difficult indoor channel conditions, such as multipath fading and shadowing, which are also critical challenges [14,15].

Nonetheless, because most services are known to be location-aware, the rapid growth of the time people spend indoors necessitates an efficient and robust new approach to localization systems using the existing wireless communication infrastructure. Furthermore, as most IoT applications take place indoors, the emergence of the IoT as a ubiquitous paradigm over the last two decades has positively influenced the demand for IPSs. Along with this, various indoor positioning technologies have been proposed in the literature to locate both users and mobile devices regardless of the complexity of the indoor environment, such as FM radio [18], ultrasound [19], light [20], magnetic field [21], and radio frequency technologies (Wi-Fi [22], radio frequency identification (RFID) [23], Bluetooth [24], Ultra-Wide Band (UWB) [25], Zigbee [26]), pedestrian inertial sensing [27], infrared [28], and many others. Due to the rapidly growing commercial interest in indoor location-based services (ILBSs) and the increasing demand for IPSs in the IoT, location systems need to be efficient, robust, and consider both the cost and practicality of their implementation. However, existing research on IPSs is constrained in scope, often necessitating the selection of specific measurement techniques from a wide range of options. Additionally, the technologies employed for indoor positioning applications rely only on certain dimensions of the system model, despite the fact that the performance of the positioning system is influenced by a multitude of interconnected factors.

On the other hand, several surveys in the literature focus on existing IPSs, examining the techniques and technologies used for the indoor positioning of both users and mobile devices. For instance, a review in [29] explores recent advancements in Wi-Fi fingerprint localization, focusing on two key areas: advanced localization techniques and efficient system deployment. It highlights how temporal and spatial signal patterns, user cooperation, and motion sensors enhance localization accuracy [29]. Additionally, the review emphasizes recent approaches for efficient system deployment, such as reducing the cost of offline data collection, adapting to fingerprint changes, calibrating heterogeneous devices for signal acquisition, and improving energy efficiency in smartphones. Similarly, the authors of [30] provide a survey of indoor localization technologies, methods, and performance, with a particular focus on recent advances in Wi-Fi fingerprinting, discussing various approaches and limitations. Furthermore, the survey in [31] offers a comprehensive overview of wireless-network-based localization, tracking, and navigation technologies, with a focus on range-free localization techniques, cellular-network-based user mobility estimation, and the latest smartphone-based approaches for simultaneous indoor localization and mapping (SLAM).

In [32], a unified framework for indoor positioning by fusion was proposed to examine the state of the art in fusion-based indoor positioning (FBIP), focusing on what should be fused and how. A review paper [33] also provides a comprehensive overview of positioning technology using channel state information (CSI), focusing on two approaches: geometry-based positioning and fingerprint-based positioning. A survey paper [34] identifies the specific requirements of an IPS for emergency responders and guides localization techniques and methods, highlighting the advantages and disadvantages of their use. In general, IPSs have been explored over the past two decades to provide users with a wide range of services, from simple LBSs, such as commercial tracking services, complex venue management, self-driving cars, and security monitoring, to the more dangerous and time-critical task of locating and tracking a wildland firefighter [10,11,12,13,32,33,34]. We have noted that several surveys on IPSs have provided an overview of approaches in specific areas and also described details of recent advances and developments in indoor positioning systems aimed at the rapid expansion of IoT applications, and these can be considered the strong points of the surveys available in the literature.

Nevertheless, there is still a great need among industry, commerce, practitioners, engineers, and researchers for an up-to-date comprehensive review work on theories and methods of IPSs and their impact on IoT, as the currently proposed solutions are mainly related to a specific use case that cannot provide the required positioning performance expected from important applications or does not cover all dimensions of the criterion, mainly for two reasons: (a) most of the existing surveys are either limited in scope or do not represent the holistic phenomena of indoor positioning, either in theory or in methods; and (b) the existing surveys do not provide an in-depth and detailed discussion on the theories and methods of both measurement techniques and technologies intended for indoor positioning systems to propose better alternative options in both techniques and technologies. The contributions in this survey paper are as follows:We investigate the various measurement techniques and technological solutions used to address complicated indoor scenarios and present a comprehensive overview of the theories and methods of indoor positioning systems, focusing on the measurement techniques, technologies, and methods used for indoor positioning systems in general.We briefly discuss and explain how the impact of IoT as a pervasive paradigm could also open up a wide range of driving factors for IPSs, including the opportunities for and challenges of using IoT infrastructure for indoor location purposes.We aim to introduce the reader to some of the current location systems and evaluate these systems using multidimensional matrices, considering both efficiency and cost, as well as practicality, in line with the evaluation metrics described in the literature.We have also created a general framework for indoor location systems that allows us to provide a condensed overview of recent advances and developments in indoor positioning. This framework also serves researchers and practitioners to better understand the general milestones in the state of the art and identify challenges as future research directions.We briefly present the approaches and methods of transfer learning from the perspective of data and models to improve the performance of indoor positioning.We propose some feature engineering techniques that can be used as countermeasures to improve positioning performance by addressing irrelevant features that may affect the overall performance of the system model.We provide an overview of ensemble learning and explain how it can be effectively used to improve the overall estimation and positioning accuracy in IPSs.We explain in detail how data fusion techniques and multisensory technology can be used to minimize the real challenges in various applications of IPSs.

To ensure the validity and rigor of this paper, we employed a systematic methodology encompassing the searching, screening, extracting, summarizing, and discussing of information from relevant sources. A comprehensive literature search was conducted across multiple databases, primarily focusing on IEEE (IEEE Xplore Digital Library), *Sensors* (MDPI), ACM (ACM Digital Library), and Wiley (Wiley Online Library), all of which are SCI-indexed. These databases were selected for their extensive coverage of peer-reviewed literature in engineering, technology, and applied sciences, particularly in the field of IPSs, facilitating the identification of influential works. We established clear inclusion criteria, focusing on peer-reviewed articles published within the last ten years related to IPSs (for almost all the references we cited), while excluding non-English articles and non-peer-reviewed literature. The initial search results were screened based on titles and abstracts, with full-text reviews confirming relevance to our study objectives. Systematic data extraction was performed to gather key findings, methodologies, and pertinent statistics. The extracted data were thematically organized to summarize trends and significant findings, enhancing the understanding of the existing research landscape. Finally, the findings were discussed in relation to the literature, identifying areas for future research and implications for practice. This systematic approach, supported by SCI-indexed sources, ensures the quality and relevance of the materials included in our review.

The remainder of this survey is structured as follows: Section 2 presents the fundamental theories and methods of IPSs, focusing on measurement techniques and network technologies. It also offers a comprehensive overview of recent advancements in IPSs, targeting researchers and practitioners, and providing a general framework for understanding key milestones in the state of the art. Section 3 discusses the impact of the IoT on IPSs, highlighting associated opportunities and challenges. Section 4 reviews data fusion techniques used to address indoor signal fluctuations, drawing on various case studies from the literature, experimental fieldwork, and our own experiences with different scenarios. Section 5 covers transfer learning techniques and methods for managing indoor signal pattern variations, based on multiple use cases. Section 6 examines architectural considerations, challenges, and future directions for indoor location system deployment, offering insights for future research. Finally, conclusions are presented in Section 7.

## 2. Fundamental Theories and Methods of IPSs

This section provides an overview of various techniques and technologies applied to IPSs, focusing on the system modeling of IP-based signal features, with an emphasis on measurement techniques, technologies, and methods addressing the indoor positioning problem (IPP). Additionally, we offer a comprehensive description of the key positioning techniques used for both users and mobile devices, emphasizing network technologies, positioning architectures, and systems. We also discuss potential architectural design metrics for position estimation and the underlying challenges that limit the application of these signal features in solving the IPP.

### 2.1. Indoor Positioning Overview

With the advancement of LBSs, location applications have garnered significant attention. Various location systems play a critical role in LBSs, enabling tracking and navigation of user positions. These systems are generally classified into three categories: (i) indoor location systems, (ii) wide-area location systems based on cellular networks, and (iii) global location systems. Indoor positioning technologies are further divided into two groups based on signal type: (i) radio-based signals and (ii) non-radio-based signals. Radio-based technologies include Wi-Fi, RFID, Bluetooth, Zigbee, and UWB, while non-radio technologies involve ultrasound, infrared, and geomagnetic fields. Three main algorithms [35] are commonly applied in IPSs: (a) triangulation, (b) proximity-based estimation, and (c) scene analysis. Triangulation estimates a target’s location using geometric principles, typically employing lateration (distance measurements) and angulation (angle measurements) from multiple reference points. Proximity-based estimation determines location based on the nearest known points, often using a network of antennas [36]. However, this method can only provide an estimate based on the strongest signal, which limits its accuracy [37]. Scene analysis relies on received signal strength (RSS) from Wi-Fi networks. It involves two phases: the training phase, in which RSS data are collected and stored in a radio map, and the online phase, in which this model predicts the user’s location based on real-time measurements [38,39,40]. To effectively implement these technologies at scale, the system architecture must be designed according to specific application needs. The following sections will detail three architectural frameworks for location services based on object accountability for processing the position:Mobile Device-Based IPS (MDBIP)

The architecture of an IPS is shaped by design factors, including specific use cases or applications, as well as the positioning topologies of how data are collected, sensed, and processed. For navigation applications, the Mobile-Device-Based Positioning (MDBP) architecture is preferred. In this setup, mobile users receive signals from anchor nodes to determine their location within the indoor environment. The MDBP has its trade-offs: localization processing occurs on the user’s device, which does not impact other users or anchor nodes. While this approach places the processing burden on mobile devices, it enhances system scalability, enabling the simultaneous tracking of many users across a larger area. Figure 1a illustrates the architecture for Mobile-Device-Based IPS (MDBIP).

2.Anchor-Based IPS (ANBIP)

For user location, Anchor-Based Positioning (ANBP) is often preferred. In this approach, anchor nodes receive signals from mobile users to facilitate positioning and related services. The choice of positioning topology directly influences the system’s architectural design based on application goals. Unlike the MDBP architecture, ANBP involves a server-side localization process, in which the server handles multiple user requests simultaneously in an indoor environment. While this architecture is advantageous for maintaining the primary communication function of wireless technology, it can introduce latency due to the server managing concurrent requests. Figure 1b illustrates the architecture for ANBIP.

3.Network-Based IPS (NWBIP)

The network-based architecture of positioning is basically the underlying network technology that is accountable to both measurements and processing localization in an indoor environment. For instance, if the BLE (Bluetooth LE) is the network technology installed in a building, the Bluetooth sniffers detect advertisement packets from Bluetooth devices. The advantage of NWBIP is that the positioning accuracy may increase when a device is detected by several sniffers. On the other hand, this architecture has its trade-offs, such as the cost involved due to the need to install a large number of sniffers and the power consumption required to enable network connectivity [41].

### 2.2. Positioning Techniques

Positioning techniques involve various signal metrics used to localize users or devices in complex indoor environments. Key methods include Time of Arrival (TOA) [42], Angle of Arrival (AOA) [43], Received Signal Strength Indicator (RSSI) [44], Channel State Information (CSI) [45], Phase of Arrival (PoA) [46], Time Difference of Arrival (TDoA) [47], Time of Flight (ToF) [48], Return Time of Flight (RToF) [49], and Pedestrian Dead Reckoning (PDR) [50]. Hybrid techniques that combine these metrics have been developed to enhance positioning performance. Additionally, derived features, such as Signal Strength Difference (SSD) [51], Hyperbolic Location Fingerprint (HLF) [52], Difference of Fingerprint (DIFF) [53], Signal-Subspace Matching (SSM) [54], Power Delay Doppler Profile (PDDP) [55], and Fused Group of Fingerprints (FGoF) [56], further improve accuracy in indoor settings.

Received Signal Strength Indicator (RSSI)

RSS values are widely used for indoor positioning, especially in areas where GPS signals are weak or blocked. These measurements can be obtained from various network technologies, including WLAN, BLE, and GSM [57,58,59]. The RSS indicator (in dBm) measures the power level of the received signal and is often used to estimate the distance between the transmitter and receiver [60]. There are two main approaches to RSS-based indoor positioning: the model-based approach and the fingerprinting approach. The model-based approach relies on the relationship between the RSS values and the distance to determine the target’s location [60], estimating the position of a mobile device by calculating the distance based on the RSS value, as described in Equation (1).
(1)$$PL={\left(P_{Tx}\right)}_{dBm}-{\left(P_{Rx}\right)}_{dBm}=PL_{0}+10n\log_{10}^{\left(\frac{d}{d_{0}}\right)}+s_{\sigma }$$
where PL, ${\left(P_{Tx}\right)}_{dBm}$, and $P_{RxdBm}$ represent the total path loss at distance d, the transmitted power, and the received power in dBm, respectively. Similarly, $PL_{0}$ denotes the path loss at the reference distance $d_{0}$, which is calculated based on the free space path loss model. Moreover, the random noise term follows the log-normal distribution with a mean of 0 and variance $\sigma^{2}$, that is, $S_{\sigma }∼\log N\left(0,\sigma^{2}\right),$ representing the attenuation in decibels caused by complex indoor environments. Although model-based RSS approaches for indoor positioning are cost-effective and easy to implement, they suffer from significant limitations due to the large fluctuations in RSS values in complex indoor environments [15,61,62]. To address these fluctuations, techniques like fingerprint cluster aggregation and filtering have been introduced [63,64]. However, achieving high positioning accuracy with the model-based RSS approach requires a large dataset of training instances and complex algorithms, which increases computational demands and complexity. Conversely, the RSS-based fingerprinting method is the most widely adopted indoor localization technique [57]. Despite its popularity, its performance can be affected by temporal variations in RSS, device heterogeneity, and fingerprint duplication, as shown in Figure 2 [65]. The figure demonstrates that RSS measurements from a Wi-Fi access point exhibit a wide range of signal values at various reference points. When signal fluctuations increase, the accuracy of “signal-to-location” mapping weakens, and outliers further degrade positioning estimates. At target 1, for example, there are extreme signal fluctuations. Nevertheless, RSS-based fingerprinting is favored for indoor localization because it does not rely on geometric estimates and typically outperforms model-based methods in complex environments [61,62,63,64].

Additionally, a SWOT analysis, shown in Table 1, is used to provide a clear understanding of RSS measurements. This helps readers, engineers, researchers, and practitioners gain insights, serving as a foundation for future research in indoor positioning applications.

2.Channel State Information (CSI)

CSI is a feature of modern wireless communication technology that emerged with the release of the Intel Wi-Fi Link 5300 Network Interface Card (NIC), which uses OFDM-MIMO techniques [66]. CSI is now considered a more advanced wireless channel metric for IPSs, offering higher data throughput compared to RSS [67,68]. In wireless communications, CSI describes how a signal travels through the channel between the transmitter and receiver, providing more detailed information about the channel’s properties than RSSI [66,67,68]. Thus, the received signal power after passing the multipath channel of the modern wireless communication systems can be represented as follows:(2)Y=HX+ϕ
where *Y*, *X*, H, and ϕ represent the received signal, the transmitted signal vector, the channel matrix, and the AGWN (additive Gaussian white noise), respectively, such that $\phi ∼N\left(0,\sigma^{2}\right)$. Moreover, each group of CSIs is a complex number, which can be represented as amplitude (magnitude) and the phase of an OFDM subcarrier, as
(3)$$H_{m}=\left|H_{m}\right|e^{k∠H_{m}}$$
and $\left|H_{m}\right|$ and $∠H_{m}$ represent the amplitude and phase of m subcarriers, respectively. With the introduction of the CSItool, the multipath channel effect can be analyzed using CSI features extracted from Wi-Fi networks. As a result, CSI measurements are becoming a viable alternative for indoor positioning fingerprints. Recent studies have shown that CSI-based fingerprints provide promising and robust performance in dynamic indoor environments [69,70,71]. This is due to CSI’s ability to capture detailed amplitude and phase information, which better represents the multipath effect, making it more stable and reliable compared to RSS, which only measures the combined amplitude of all subcarriers.

Several object detection schemes using WLAN CSI fingerprints have been proposed [72,73,74]. In OFDM systems, the CSI at each subcarrier can capture a target’s behavior through amplitude and phase fluctuations in a frequency-selective fading channel. MIMO technology in CSI further enhances positioning by providing high-dimensional features, although this comes with the trade-off of increased computational complexity. However, high-dimensional features alone may not always improve localization accuracy, as redundant features can degrade model performance. Identifying the most significant predictors is crucial, as shown in Figure 3, where classifiers trained on different feature sets (dimensions of 41 and 15) have significantly lower root mean square errors, reducing mis-localization rates by 32% and 25% [65]. A comparative study of CSI and RSSI revealed several advantages of CSI: (a) it effectively handles multipath propagation, (b) it has strong stability in static environments and relative resilience to environmental changes, and (c) it reduces radio interference from carrier frequency signals [75,76,77]. However, CSI-based indoor positioning systems still face challenges in managing severe dynamic range fluctuations, radio interference, multipath effects, and random noise, even in stationary environments [69,78,79]. These issues must be addressed in future research. Table 2 presents a SWOT analysis of CSI-based fingerprinting for indoor positioning systems.

3.Time of Arrival (TOA)

In the TOA method (i.e., time of flight (ToF)), the signal propagation time and the known signal velocity are the two elements used to estimate the distance between the sender and receiver of a signal [80]. In the TOA-based ranging technique, both the transmitter $T_{X}$ and receivers $R_{X}=\left[R_{1},R_{2},\ldots ,R_{n}\right]$ are considered to be synchronized. Suppose a wireless network is given as depicted in Figure 4, where there are n numbers of anchor nodes (ANs), $\hat{p}={\left[\hat{x},\hat{y}\right]}^{T}$ is the estimate of the mobile terminal (MT) location, $p_{i}={\left[x_{i},y_{i}\right]}^{T}$ is the position of the $i^{th}$ AN (i=1,2,…,n), and ${\hat{d}}_{i}$ is the measured distance between the MT and the $i^{th}$ AN, commonly modeled under the LOS environment as follows:(4)$${\hat{d}}_{i}=d_{i}+\varepsilon_{i}=c\Delta t_{i}$$
where $\Delta t_{i}=\left(t_{i}-t_{0}\right)$ is the TOA of the signal at the $i^{th}$ AN, *c* is the speed of light, $d_{i}$ is the absolute distance between the MT and the $i^{th}$ AN, and $\varepsilon_{i}∼N\left(0,\sigma_{i}^{2}\right)$ is the additive white Gaussian noise (AWGN) with variance $\sigma_{i}^{2}$. However, the TOA fingerprint is dynamic in indoor environments due to the effect of multipath propagation, such as NLOS and random channel noise during measurements. Hence, ${\hat{d}}_{i}$ is the measured distance between the MT and the $i^{th}$ AN, which needs to be remodeled under the NLOS environment [81] as
(5)$${\hat{d}}_{i}=d_{i}+b_{i}+\varepsilon_{i}=c\Delta t_{i}$$
where the term $b_{i}$ in Equation (4) is a positive distance bias introduced due to the NLOS scenario. Thus, ${\hat{d}}_{i}$ is the measured distance between the MT and the $i^{th}$ AN, and it is calculated as follows:(6)$${\left\|d_{i}\right\|}_{2}^{2}={\left(x_{i}-\hat{x}\right)}^{2}+{\left(y_{i}-\hat{y}\right)}^{2}$$

For the NLOS scenario, the bias term $\varepsilon_{i}$ has been modeled in various ways in the literature, including but not limited to exponential distribution [82], uniform distribution [83], Gaussian distribution [84], and constant with a time window [85], or in an empirical model from measurements [86]. Based on the TOA ranging technique, as shown in Figure 4, the target or mobile device is located in the circle centered on the *i*-th base station (at the intersection of three circles), with a radius given in Equation (4), where c is the speed of light in noise-free space [87]. Several localization techniques based on TOA measurements, such as the Taylor algorithm [81], the Chan algorithm [88], the least-squares method (LS) [89], and weighted least squares (WLS) [90], have been proposed in the literature to determine the position of the mobile terminal from (5) in both LOS and NLOS scenarios. In addition, TOA-based positioning can be categorized into two types, depending on whether positioning is performed on the receiver or transmitter side. In the One-Way Time of Arrival (1W-TOA) method, the one-way propagation time is measured, which is the time difference between when the signal is transmitted and when it arrives. This method requires very precise synchronization between the clocks of both the transmitter and receiver. In contrast, Bidirectional Time of Arrival (2W-TOA), also known as Return Time of Flight (RToF), measures the round-trip time of a signal and is preferred over 1W-TOA for indoor positioning. Table 3 presents a SWOT analysis of TOA-based fingerprinting in indoor location systems.

4.Time Difference of Arrival (TDOA)

The TDOA approach uses two signals that travel with different velocities to estimate the position of a user or mobile device, unlike the TOA ranging technique that uses the absolute signal propagation time [81]. Indoor positioning based TDOA fingerprint needs at least three reference nodes or transmitters to estimate the receiver’s location as the intersection of two or more hyperboloids are sufficient to localize the target or mobile device [91] using linear regression or with the help of Taylor-series expansion to linearize the hyperbolic non-linear model [92]. Thus, the receiver’s location on the hyperboloid can be given as:(7)$${\hat{d}}_{ij}=\left(v_{j}-v_{i}\right)\times \Delta t$$
where $\Delta t=\left(t_{4}-t_{2}-t_{wait}\right)$ and $t_{wait}=t_{3}-t_{1}$, $t_{1}$, $t_{2}$, $t_{3}$ and $t_{4}$ represents the signal’s sending time of the first transmitter, signal’s receiving time at the first receiver, signal’s sending time of the second transmitter, and signal’s receiving time at the second receiver respectively. The main challenge in indoor positioning using TDOA measurements is that the model’s parameters are non-convex and non-linear, as the target’s location is estimated from the intersection of two hyperbolic curves [91]. Using the maximum likelihood estimation (MLE) method directly to solve this model is not ideal due to its high computational cost [92]. To overcome this, a weighted least squares method was proposed, which transforms the TDOA hyperbolic model into a set of linear equations by introducing two noise parameters. This provides a closed-form solution and achieves high positioning accuracy [81]. Several other methods, including the Taylor algorithm, Chan algorithm, and robust least squares with semidefinite relaxation, have also been developed to address the non-convex and non-linear nature of TDOA-based positioning [93]. TDOA-based indoor positioning has shown high accuracy in LOS conditions. Additionally, it does not require extra hardware for clock synchronization between the transmitter and receiver, making it preferable to TOA-based positioning [92,94,95]. However, in complex indoor environments with multipath effects or when LOS is unavailable, TDOA’s performance is less reliable, especially in NLOS conditions or dynamic environments. Table 4 presents a SWOT analysis of TDOA-based positioning.

5.Angle of Arrival (AOA)

AOA is used to determine the direction of a signal’s propagation, typically by using an antenna array on the receiver side and applying triangulation, as shown in Figure 5 [96]. In AOA-based indoor positioning, MIMO antenna groups can measure the signal’s arrival angle in various ways, depending on the system’s architecture [97]. For example, the signal emitted from the mobile device or user reaches the antenna groups with an angle β and the time for the signal from the transmitter to reach antenna 2 is longer than that to antenna 1. Therefore, the time delay is represented as t_delay_ and the AOA can be estimated by the triangulation method as $\beta =\arcsin\frac{t_{delay}\times c}{d}$, where c is the speed of light. AOA refers to the angle between a signal’s propagation direction and a reference direction, often used in array signal processing [98]. In indoor positioning, AOA measurements can be performed with two anchor nodes to determine a device or user’s location in a 2D space, offering more flexibility compared to systems based on RSS or TOA techniques [32]. AOA-based systems perform better at short distances between transceivers. However, the performance decreases with distance, and additional hardware and device calibration are required, which limits its practicality compared to RSS-based systems [91,96,97,98]. Furthermore, AOA-based positioning is significantly affected by complex indoor environments, such as NLOS and multipath effects. Table 5 presents a SWOT analysis of AOA-based indoor positioning. To improve accuracy and reduce the impact of multipath propagation, methods like Multiple Signal Classification (MUSIC) [99] and Estimation of Signal Parameters via Rotational Invariance Techniques (ESPRIT) [100] have been proposed.

6.Hybrid signal features

Several single-signal-based methods have been developed to address indoor positioning, but they face challenges such as signal variability and lack robustness in dynamic environments. These limitations arise because (i) measurements rely on single fingerprints, (ii) updating the fingerprint database frequently is impractical, and (iii) collecting labeled data is time-consuming and costly. To overcome these issues, hybrid localization systems [8,12,39] have been introduced. By combining two or more signal measurement techniques, hybrid systems improve positioning accuracy and ensure service availability, even during system failures, as they are not standalone systems. However, they are costly to implement, and their high computational demands limit widespread use.

### 2.3. Indoor Positioning Principle Algorithms

In this section, we discuss the core algorithms using IPSs, which serve as the foundations for many modern techniques in the field. Indoor positioning techniques can be broadly categorized into dead reckoning positioning, fingerprinting and other forms of pattern-matching positioning, and spatial geometric relationship positioning. These algorithms are integral to understanding how data from various sensors and signals are processed to estimate a subject’s location.

1.Pedestrian Dead Reckoning (PDR)

In PDR-based indoor positioning systems, three important types of information are required to estimate the current position of a target, namely, stride length, walking direction, and previous estimated position [101]. Thus, one can estimate the current position of a target based on PDR as follows:(8)$${\hat{p}}_{i}={\hat{p}}_{i-1}+d_{i}$$
where ${\hat{p}}_{i-1}$ and ${\hat{p}}_{i}$ are the location estimates before and after the kth step, respectively, and $d_{i}$ is the estimated displacement vector for the $i^{th}$ step, which is given as $d_{i}={\left[\alpha_{i}\cos\beta_{i},\alpha_{i}\sin\beta_{i}\right]}^{T}.$ $\alpha_{i}$ and $\beta_{i}$ are the stride length and heading orientation for the $i^{th}$ step, respectively. The detailed implementation of the PDR algorithm is beyond the scope of this paper. For a practical step-based PDR system, which includes key operations like orientation and projection, refer to [101]. A major issue with PDR-based IPSs is that errors in estimated displacement accumulate rapidly due to the double integration of noisy sensor readings [102,103]. To address this, an algorithm called Zero Velocity Update (ZUPT) was proposed [104], which helps to reduce cumulative tracking errors over long distances, although it requires additional hardware for a sensor module. Table 6 presents a SWOT analysis of PDR-based indoor positioning. However, the ZUPT algorithm is limited to specific users, such as rescue workers and firefighters, due to the inconvenience of wearing a foot-mounted sensor module [104].

2.Fingerprinting and Pattern Matching Positioning

Fingerprinting is a widely used technique that relies on matching the current signal characteristics to a pre-recorded database of signal measurements, or “fingerprints”. Wi-Fi, Bluetooth, or other signals are often used for this purpose. The accuracy of fingerprinting largely depends on the density of the pre-collected database and the stability of the signals in the environment. Pattern matching techniques, in general, compare real-time sensor data with known patterns to estimate the position [57,58,59,62,63,64,65]. Figure 6 illustrates the overall workflow of a fingerprinting-based indoor positioning system. In the training phase, signals from multiple access points (AP1, AP2, …, APm) are collected at known reference points (RPs) with varying signal strengths, depicted by different colors. The collected signals are stored in a radio map or fingerprinting database. During the localization phase, signals are measured at an unknown location and compared with the pre-stored fingerprints using a fingerprint matching algorithm. The algorithm determines the closest match in the database, which is then used to estimate the target’s location.

3.Spatial Geometric Relationship Positioning

This method involves geometric algorithms, such as trilateration or triangulation, to calculate the subject’s position by measuring the distances or angles from known reference points. These techniques work similarly to the GPS, but they are adapted for indoor environments using radio signals, UWB, or other proximity sensors. Geometric methods are often used in conjunction with other algorithms to improve accuracy. Table 7, below, provides a detailed comparison of the two principal approaches for RSS-based IPSs: the model-based approach and the fingerprinting approach [60,61,62,63]. The comparison is structured around key criteria, including the underlying methodology, accuracy, required setup effort, and adaptability to environmental changes. This analysis highlights the distinct advantages and limitations of each approach, offering critical insights for their application in varied indoor localization contexts.

### 2.4. Positioning Technologies

This section provides an overview of various indoor positioning technologies, along with brief descriptions of the most commonly used systems in the literature. These positioning systems utilize different network technologies to estimate the location of a target. Key technologies include Wi-Fi, BLE, RFID, geomagnetic systems, UWB, inertial navigation systems (INS), ZigBee, visible light, and ultrasound-based positioning systems.

1.WLAN-Based Indoor Positioning System

WLAN-based positioning systems are widely used due to their accessibility and minimal infrastructure investment. They typically employ two main approaches: (a) range-based and (b) fingerprint-based. The range-based approach relies on precise geometric parameters derived from signal features such as RSS [21], TOA [22], AOA [23], and TDOA [24] for accurate positioning. However, indoor environments often present challenges like NLOS propagation, multipath effects, and dynamic changes, which hinder the accuracy of this method [61,62]. In contrast, the fingerprint-based approach has gained popularity for indoor localization, as it does not depend on geometric parameter estimation and typically outperforms range-based methods in complex settings [63,64]. This method utilizes RSS values from Wi-Fi networks to establish a relationship between transceivers and measure localization accuracy based on distances to available Wi-Fi access points [38,39,40]. RSS fingerprints are collected at various grid points to create a radio map, which is used to train a predictive model during the training phase. In the testing phase, this model infers the location of a mobile device based on new measurements [38,39,40]. Despite its advantages, such as universal availability, privacy protection, and low implementation costs, Wi-Fi fingerprinting faces challenges, including the high expense of creating wireless maps [25,26,27]. Some strategies, like crowd sourcing [28] and simultaneous Wi-Fi localization and mapping, have been proposed to alleviate the effort and time required for radio map generation [29]. However, crowd sourcing relies on user participation and may yield low accuracy, while simultaneous mapping involves significant computational demands. Additionally, existing Wi-Fi networks are primarily designed for communication, not positioning, necessitating the development of robust algorithms for improved localization performance. Wi-Fi technologies, based on IEEE 802.11 standards [25,26,27], operate on ISM bands and are essential for indoor Internet access, requiring considerations of bandwidth to enhance data throughput and mitigate channel interference, which impacts location accuracy. Combining various signal features, such as RSS, CSI, AOA, and TOF/TOA, could further enhance Wi-Fi-based location services.

2.Bluetooth-Based Indoor Positioning System

BLE technology offers significantly lower power consumption and lower bit rates compared to Wi-Fi, making it ideal for short-distance data transmission [105]. This efficiency has led to the development of various IoT devices, such as beacons, which are small, cost-effective devices that transmit information packets to nearby BLE-enabled devices at regular intervals [106]. A study [107] compared Wi-Fi, BLE, Zigbee, and LoRaWAN for indoor localization, utilizing three transmitting nodes and a single receiver based on the trilateration method to estimate location. While Wi-Fi IPSs provide the highest accuracy, BLE has emerged as a promising option for short-range networks due to its low power consumption and minimal costs [107]. For instance, Gaussian filters have been used to enhance fingerprint-based IPS with BLE beacons [108]. Research has also explored BLE performance in typical indoor scenarios, such as conference rooms and showrooms, analyzing factors like direction, sampling rate, distance, time, sample size, and obstacles [109]. The results indicated that BLE’s lifespan decreases slightly slower than Wi-Fi, which has a high-power consumption of about 2.5 h [109]. Additionally, a hybrid approach combining fingerprinting and trilateration methods has been proposed for enhancing IPSs’ accuracy [110]. BLE fingerprinting technology has also been investigated for target detection at various beacon densities [111].

3.RFID-Based Indoor Positioning System

RFID-tag-based positioning systems are recognized for their high accuracy, moderate cost, and ability to penetrate obstacles. These systems consist of two main components: RFID tags (active and passive) and RFID readers [112]. Active RFID tags are battery-powered and can transmit signals independently, while passive tags rely on an external source to initiate signal transmission [112]. RFID technology is favored for its advantages, including high data rates, functionality in NLOS environments, strong security, cost-effectiveness, and compact design [113,114]. Localization can be achieved through two methods: (a) centralized localization, in which the tag attached to the target object determines its position, and (b) distributed scalable localization, in which the tag reads its coordinates from nearby readers with known locations. RFID localization can be categorized into reader localization and tag localization. For instance, a reader-based method for mobile robots was proposed in [115], while the LANDMARC technique utilizes reference tags at known locations to aid localization [116]. The accuracy of this system is influenced by the placement and density of the reference tags used [116].

4.UWB-Based Indoor Positioning System

UWB is a wireless communication technology that uses non-sine-wave narrow pulses, ranging from nanoseconds to microseconds, to transmit data [117]. UWB IPSs are known for their high-power penetration, large bandwidth, and centimeter-level positioning accuracy, which can even reach millimetric precision under LOS conditions [118,119]. However, practical challenges such as multipath fading and NLOS conditions can attenuate signals, leading to inaccuracies in distance measurements. UWB technology employs low-power spectral density and narrow pulse signals, allowing for high temporal resolution and strong barrier penetration. Its features make UWB an attractive wireless solution for indoor positioning [120]. While UWB can achieve centimeter-level accuracy using TOA measurements and trilateration, the requirement for additional hardware and higher power consumption limits its cost-effectiveness compared to Wi-Fi and BLE [121]. UWB is primarily suited to short-distance applications and high-speed data transmissions, such as radar, object tracking, and medical monitoring, due to power consumption constraints [22]. The accuracy of UWB positioning is influenced by the number and placement of anchors, as well as LOS coverage [122]. Integrating trilateration and fingerprinting methods has been proposed to lower costs in UWB localization [122]. A hybrid approach combining UWB and inertial navigation has shown improved indoor positioning performance, with experimental results indicating a 45% reduction in mean error compared to UWB alone [123].

5.Inertial Navigation Systems (INSs)-based Indoor Positioning System

Accurate location information is crucial for various applications, including military operations and indoor emergency services. A range of navigation and positioning technologies exists, from GPS- to radio-network-based systems, but these often require local infrastructure, which can be compromised in emergencies [124]. INSs provide an autonomous solution using data from Inertial Measurement Units (IMUs), which consist of accelerometers and gyroscopes. INSs can calculate the system’s current speed, position, and orientation from a known starting point. However, errors in these measurements, such as gyroscope drift and scale factor inaccuracies, tend to accumulate over time [125,126]. To enhance positioning accuracy, a new IPS combines UWB-sensor-based and IMU-based solutions. This integration aims to mitigate errors introduced by the INS [127]. Additionally, a distributed system has been developed for personal positioning, utilizing an IMU connected to a wearable radio and a server. The IMU estimates stride length and direction, which are then processed by the server using map matching with particle filters. This distributed architecture allows for low user-side computing requirements, resulting in extended uptime and reduced power consumption [128].

6.Cellular-Network-Based Indoor Positioning System

In a cellular network, base stations are distributed over a region, with each covering a small area, known as a cell, as illustrated in Figure 7, with Global System for Mobile Communications (GSM) and Code Division Multiple Access (CDMA) technologies [129]. The literature indicates that cellular-network-based location methods can estimate the locations of mobile terminals in various environments, both indoor and outdoor, although they generally lack the accuracy of GPS [130,131]. GPS often struggles to provide reliable location estimates in dense urban and indoor settings due to signal fading or obstruction. To address the limitations of indoor cellular communication, a hybrid positioning approach has been proposed, combining measurements from cellular networks and GPS [132,133]. For example, a cooperative network framework integrates cellular networks with other systems, such as WLAN and FM, to improve indoor positioning [134]. Another study introduced a hybrid localization method using an extended Kalman filter (EKF), which combines round-trip time (RTT) and RSS measurements from GSM with pseudo-range data from GPS for tracking mobile terminal motion [135]. Cellular-network-based location methods can utilize various measurement techniques, including RSS, TOA, RTT, LoRa, and TDOA [135,136,137]. Signal interference scenarios in mobile networks are depicted in Figure 7.

7.ZigBee-Based Indoor Positioning System

A ZigBee-based tracking system is designed for energy-efficient applications, prioritizing low data rates, low cost, and long battery life. In one study, a mobile patient monitoring network using ADV (Ad Hoc On-Demand Distance Vector) and DSR (Dynamic Source Routing) algorithms was analyzed, showing that the ADV protocol was best suited for medical surveillance due to its superior performance in terms of delay reduction and efficiency [138]. Another study proposed an indoor positioning algorithm based on ZigBee, utilizing an exponentially weighted moving average to smooth signal propagation and improve system performance [139]. Since radio signals fluctuate over time, updating the radio map is crucial for adapting to changes in the environment [140]. Pre-processing techniques, such as adjusting reference node positions, calibrating RSSI values, reducing output power, and deactivating irrelevant wireless nodes, were used to mitigate signal jitter and multipath effects [140].

8.Visible Light Indoor Positioning System

Light-emitting diodes (LEDs) offer key advantages like high bandwidth [141], energy efficiency [142], durability [142], and cost-effectiveness [143], making them promising alternatives to radio frequency in positioning systems. VLIP uses LEDs for illumination and photodiodes (PDs) as receivers [144]. However, existing VLIP-based methods on smartphones often face high computational costs for image processing, limiting their positioning accuracy. Despite this, VLIP is viewed as a cost-effective solution for real-time indoor positioning [145]. One study proposed a positioning system using LED illumination and image sensors, showing improved accuracy and directional measurement [146]. Experimental results demonstrated that this system could achieve a positioning accuracy of 1.5 m by capturing different three-dimensional spatial coordinates illuminated by LEDs [147]. Another VLIP solution designed a jitter-free encoding scheme and a lightweight image processing algorithm, achieving precise navigation with an accuracy of up to 5 cm and reducing computational time to 22.7 ms for one lamp and 7 ms for two lamps [148].

Several smartphone-based VLIP system prototypes have been developed, such as Epsilon [149], Landmark [150], Luxapose [151], and PIXEL [152]. However, these systems often struggle with positioning accuracy [148,153] and speed [154]. The challenge lies in extracting information from light signals through image processing, which is computationally demanding for smartphone processors, even with recent improvements in processing power. These systems balance trade-offs between accuracy and latency. Various techniques have been proposed for positioning in VLC systems, including RSS [155], AOA [156], TOA [157], and TDOA [158]. Additionally, several algorithms have been tested for indoor positioning using visible light signals [144,145,146,147,148]. Machine learning techniques have been employed to estimate target positions based on light signals, utilizing maximum power spectral density values to create RSS fingerprints [20]. For effective indoor positioning with RSS fingerprints, at least two LEDs must be at the same height and within the LOS of the photodiode, using Lambertian light sources to estimate distance based on signal strength [159]. While VLIP systems are effective and demonstrate high positioning performance, they require direct capture of light sources, which can be impractical, as light sources may not always maintain a direct LOS [146,147,148,159].

9.Geomagnetic-Based Indoor Positioning System

The Earth’s geomagnetic field has gained attention as a promising alternative for indoor positioning, offering characteristics such as ubiquity, temporal stability, and no deployment costs. This contrasts with widely used wireless technologies like Wi-Fi and Bluetooth, which require infrastructure such as access points and beacons, along with ongoing software configurations and potential interference from other devices [160,161,162,163]. Research has shown that geomagnetic fingerprints can be effectively used for indoor location services, as the magnetic field is readily available and can be easily captured using magnetometers found in smart devices. These fingerprints can distinguish spatial variations within indoor environments [164,165]. However, the magnetic signals can be distorted by nearby electrical equipment and steel structures, affecting their reliability [166,167,168]. While the unique properties of the geomagnetic field make it suitable for indoor positioning, device standardization is necessary because different devices may yield varying readings for the same location [165,166,167]. Additionally, magnetic sequences collected along the same path show high consistency, while those from different paths exhibit significant differences [169]. Longer magnetic sequences improve positioning accuracy, as they enhance the ability to distinguish between different trajectories in indoor spaces [169].

However, magnetic sensors have a notable limitation: their readings vary based on the sensor’s orientation. In smartphones, the magnetic sensor measures the magnetic field in three dimensions, and changes in orientation can lead to different readings on each axis [170]. A significant challenge for indoor positioning using geomagnetism is the need for calibration to reduce signal variations caused by device differences. Different types of hardware can produce inconsistent magnetic values at the same location, negatively impacting positioning accuracy [170].

10.Ultrasonic-Based Indoor Positioning System

Ultrasonic-based positioning systems are primarily used in indoor robotic applications, employing ultrasonic signals with TDOA and TOA measurements for positioning estimation [171,172,173]. TDOA is often preferred because it does not require synchronization hardware between receivers and transmitters [174]. Although the Bayesian recursion approach is criticized for its high computational complexity [175,176], Sequential Monte Carlo or particle filters are commonly used for mobile robot localization. Ultrasonic sensors have gained popularity due to their simplicity and low cost for indoor positioning and navigation [177,178]. The measurements in ultrasonic positioning systems typically involve TOA, TDOA, and RSS [179,180,181]. An innovative approach using mobile phone speakers generates various tones to enhance indoor positioning accuracy [182]. Modern smartphones are equipped with multiple sensors (Wi-Fi, Bluetooth, GPS, etc.) that can facilitate positioning. Each technology has trade-offs, with Wi-Fi generally offering higher accuracy than GSM and Bluetooth in indoor environments [61,62]. However, GSM’s effectiveness declines in complex indoor settings, while Bluetooth is limited to short-range positioning [105,106,107]. RFID systems require extensive deployment to achieve high accuracy, imposing burdens on users [113,114,115,116]. UWB-based positioning systems utilize time differences in radio signal propagation but currently lack sufficient accuracy for pedestrian navigation [183]. This could be justified because radio signals require highly precise time detection and resolution due to their relatively short propagation times when compared to sound signals such as ultrasonic waves [184]. Wi-Fi technology is favored for its low cost and compatibility with existing infrastructure [38,39,40,63,64], but its accuracy can be affected by indoor complexities, typically achieving 1–2 m [185]. In contrast, unlike indoor positioning-based RF propagation signals, which are characterized by low accuracy, limited coverage, additional hardware costs, and electromagnetic interference [186], indoor positioning-based ultra-sound signals have been reported to achieve centimeter-level positioning accuracy because the speed of sound signals is significantly slower than RF, allowing for a longer time of arrival [171,172,173,174,187]. A novel method using rotating ultrasonic sensors with smartphones for processing has been proposed, improving efficiency by reducing the number of required sensors [188]. However, ultrasonic systems face challenges from noise and multipath effects in complex indoor environments, which can hinder performance [189]. Overall, while RF and ultrasonic technologies each have their trade-offs, ultrasonic systems offer benefits like low cost, reliability, scalability, and the ability to track multiple users simultaneously, making them suitable for high-precision indoor positioning [180,189,190,191,192].

11.Simultaneous Localization and Mapping (SLAM)

In this paper, we present two critical components to enrich the discussion of positioning systems: SLAM and 5G/6G network positioning. SLAM is a crucial technology in robotics, enabling a robot to navigate unknown environments by simultaneously building a map and determining its location within that map [193,194,195,196,197]. Over the past decades, SLAM has evolved from a theoretical problem into a practical solution applicable in various domains, such as indoor navigation, autonomous vehicles, underwater exploration, and aerial systems [193,194]. SLAM is typically formulated as a probabilistic estimation problem, in which the joint probability distribution of the robot’s position and the locations of observed landmarks is continuously updated [195]. The most widely used approaches for solving this problem are the Extended Kalman Filter (EKF-SLAM) and FastSLAM, both of which have been instrumental in addressing the non-linear and non-Gaussian nature of real-world environments [196]. Despite its success, SLAM still faces challenges, particularly in large-scale, dynamic environments, where issues such as loop closure, real-time processing, and data association errors require ongoing research and improvement [197]. Practical applications span various domains, including autonomous vehicles and drones, highlighting SLAM’s critical role in advancing robotic autonomy and navigation capabilities. Ongoing research continues to refine SLAM algorithms, integrating them with artificial intelligence to improve performance in complex environments [197].

In one study [198], the authors analyzed FastSLAM, a Rao–Blackwellized particle filter method for SLAM, discovering that the algorithm tends to degenerate over time, leading to overly optimistic uncertainty estimates regardless of particle count or landmark density. While FastSLAM can provide consistent short-term estimates, it faces challenges with long-term state exploration. The study suggests that, with enough particles, FastSLAM can deliver reliable non-stochastic estimates, and highlights its advantages in data association, particularly when combined with other stochastic SLAM methods, like EKF-based SLAM. Additionally, recent research [199] explored indoor mobile laser scanning (MLS) systems that apply SLAM principles to gather detailed indoor data. These portable 2D and 3D laser scanners generate point clouds that are essential for indoor navigation and architectural design, although they often produce noisy and redundant data that complicate the identification of linear edges and planes. This study proposes a semi-automatic methodology for extracting 2D and geometric information from SLAM-generated 3D point clouds to streamline indoor drawing processes. Furthermore, the research in [200] introduces Edge-SLAM, which leverages edge computing to enhance Visual-SLAM efficiency. By implementing a split architecture based on ORB-SLAM2 [201], tracking remains on the mobile device, while local mapping and loop closing are offloaded to the edge, facilitating long-term operation with limited resources while maintaining accuracy. This architecture allows for additional applications utilizing Visual-SLAM and includes a thorough performance analysis of CPU, memory, network, and power consumption, underscoring its potential for real-world use in resource-constrained settings.

12.5G/6G Network Positioning

The advent of 5G and future 6G networks introduces significant advancements in positioning accuracy and responsiveness compared to traditional cellular networks. These next-generation networks employ advanced mechanisms, such as time-based positioning signals and ultra-dense network architecture, to enhance localization capabilities [202]. The unique characteristics of 5G/6G positioning technologies allow for improved performance in dynamic environments, which is essential for applications such as autonomous navigation and smart cities. The transition from 5G-Advanced to 6G represents a significant evolution in telecommunications, particularly in network positioning capabilities. One of the defining features of 6G will be its ability to achieve high-precision positioning, with accuracy levels below one meter and update rates of a few hundred milliseconds, which is a substantial improvement over 5G’s positioning accuracy, typically within a range of 5–10 m [203]. This advancement is crucial for applications requiring real-time location data, such as autonomous driving, drone navigation, and smart agriculture [204]. In 5G, positioning is primarily focused on providing general location data within cell boundaries or large geographical areas. However, 6G will enhance this by integrating advanced technologies like Ultra-Wide Band (UWB) and advanced satellite systems, enabling precise localization in densely populated urban environments and complex indoor settings [205]. Additionally, 6G is expected to leverage artificial intelligence (AI) and machine learning to optimize positioning algorithms, improving accuracy and responsiveness in dynamic environments [206].

Another significant feature of 6G will be its ability to support high-resolution positioning through enhanced network infrastructure, including denser base station deployment and advanced signal processing techniques. This will facilitate the development of smart environments, in which positioning data can be utilized for various applications, such as precision agriculture, smart logistics, and augmented reality experiences [207]. Moreover, 6G aims to support diverse use cases through a combination of different positioning techniques, including GPS, inertial measurement units (IMUs), and machine learning algorithms. This multi-faceted approach will ensure seamless and accurate positioning services across various sectors, from healthcare to smart cities [208]. Overall, the shift to 6G will not only improve the accuracy and reliability of positioning systems but also broaden their applicability, paving the way for innovative services and enhanced user experiences. A recent paper [209] discusses three crucial services—Immersive Communications, Everything Connected, and High-Positioning—that are central to the transition from 5G to 6G in telecommunications. Supported by advancements in 3GPP Releases 17, 18, and 19, these services aim to enhance network performance. The author stresses the need for new Key Performance Indicators (KPIs) to evaluate these emerging technologies effectively. The paper also highlights key application fields likely to adopt these services in the next 3–8 years and identifies enabling technologies essential for their implementation. This analysis offers valuable insights into the future of telecommunications and the innovations shaping the next phase of mobile communications.

Below, Table 8 summarizes the key advantages and disadvantages of various indoor positioning technologies, including WLAN, BLE, RFID, UWB, INS, cellular networks, ZigBee, visible light, geomagnetic, ultrasonic systems, and 5G/6G network positioning. The table aims to provide a comprehensive overview of the strengths and limitations associated with each technology, facilitating informed decision-making for researchers and practitioners in the field of indoor localization. By highlighting performance characteristics such as accuracy, cost, energy efficiency, and operational challenges, this table serves as a valuable reference for advancing indoor positioning solutions in diverse applications.

## 3. Internet of Things and Indoor Positioning

In this section, we present an overview of the IoT, the use cases based on IoT, the impact of the IoT for indoor positioning, and applications based on IoT.

### Overview of Internet of Things

The IoT has rapidly expanded due to advancements in wireless communication and the global rollout of both 5G and 6G networks. By the end of 2020, the number of 5G users was projected to reach 220 million, reflecting a significant shift in how people connect and interact with technology [210]. The IoT facilitates applications across various sectors, including health, logistics, education, and smart cities, by enabling seamless communication and automation among diverse devices without human intervention [211,212]. Location information is crucial for IoT functionality, especially in smart homes, where the average number of connected devices per person was estimated at 4.5 in 2020 [213]. Most IoT applications occur indoors, making the integration of IPSs essential, as traditional positioning technologies like GNSS struggle with multipath effects and signal attenuation indoors. People spend approximately 70% to 90% of their time indoors [214], underscoring the importance of robust indoor positioning for applications such as LBSs and social media [215,216,217]. Despite the need for reliable indoor positioning, challenges remain due to the complexity of indoor environments. IPSs are recognized as core technologies for the IoT [218]. The IoT was projected to connect over 50 billion smart devices by 2020, enabling communication between various objects and mobile devices through technologies such as Wi-Fi, RFID, and Bluetooth [219,220,221]. Recent advancements in the IoT have led to new applications, particularly in positioning and localization. Positioning involves determining an object’s location within a specific coordinate system using mobile devices, facilitated by IPSs and LBSs. IPSs calculate a target’s location, while LBSs utilize that information to control environmental features. Thus, IPSs are vital for realizing the goals of the IoT, enabling efficient interaction between devices and users [222,223].

## 4. Data Fusion and Indoor Positioning

In this section, we present an overview of data fusion techniques and the application of these techniques in tackling the signal variations in various indoor-environment-based use cases in the literature, along with real-life experimental studies of our own experiences related to different scenarios.

### Overview of Data Fusion

Many indoor systems rely on standalone sensors, which can fail, leading to significant challenges in accuracy. The proliferation of embedded sensors in mobile devices presents an opportunity to enhance location performance by integrating multiple technologies. This integration can help address issues such as signal attenuation and reliance on single systems. Recent advancements in data fusion techniques have been proposed to mitigate signal variations and improve reliability in indoor tracking systems [224,225,226,227]. Data fusion enhances wireless sensor networks (WSNs) by increasing accuracy, reducing redundancy, and lowering energy consumption [227,228]. WSNs are characterized by high data correlation and limited energy supply, making data fusion essential for effective positioning [229,230,231]. Research indicates that fusing multiple sensor measurements significantly improves positioning accuracy compared to single-sensor systems. For instance, visual–inertial navigation systems can achieve high accuracy but suffer from cumulative errors due to sensor noise [232]. Indoor tracking systems, such as those based on PDR, are widely used but accumulate errors over time [233]. To enhance positioning systems, researchers have proposed intelligent fusion methods that combine technologies like Wi-Fi and inertial sensors, showing improved accuracy over standalone systems [234]. Other studies have integrated GPS with inertial navigation to enhance reliability during GPS outages [235]. Techniques combining TOA and RSS measurements have also been developed to improve accuracy [236]. Innovations in sensor fusion, such as using extended Kalman filters (EKF) and hierarchical localization algorithms, have been explored to improve indoor navigation for mobile robots [237,238,239]. Additionally, fusing IMU and UWB technologies has been proposed to reduce error accumulation [240]. Recent efforts have developed fusion-based systems that combine data from smartphone sensors (Wi-Fi, BLE, and PDR), demonstrating significant improvements in positioning accuracy [241]. Other approaches have focused on enhancing fingerprint-based localization methods to increase robustness and efficiency [227,242]. Advanced data fusion techniques leveraging heterogeneous knowledge transfer are designed to enhance both the accuracy and the reliability of positioning systems in dynamic indoor environments [243,244,245]. These methods show promise in overcoming the limitations of existing techniques and achieving better performance in real-world scenarios. A recent work [243] proposed a novel data fusion technique leveraging heterogeneous positive knowledge transfer to effectively represent the temporal variations of CSI in complex indoor environments, specifically targeting vehicle indoor parking scenarios. This approach aims to minimize the extensive training calibration typically required in such settings. The experimental results demonstrated that the proposed algorithm consistently delivered high positioning accuracy across various dynamic conditions, highlighting its computational robustness and efficiency. Despite these advancements, existing data fusion methodologies often fail to fully exploit the inherent relationships among multiple fingerprint functions and do not adequately utilize knowledge acquired during the offline phase [246]. To overcome these limitations, a knowledge-aided adaptive localization approach was introduced, centered around a global fusion profile, to enhance positioning performance in intricate environments [247]. As illustrated by the authors, the experimental findings reveal that this innovative approach surpasses traditional methods in both simulated and real-world scenarios [247]. In summary, various wireless technologies are available for indoor positioning. For an effective and widely deployable IPS that significantly impacts the IoT domain, it is essential to choose technologies that are readily available in smartphones and compatible with existing infrastructure.

## 5. Transfer Learning and Indoor Positioning

In this section, we present an overview of transfer learning techniques and the applications of these techniques in tackling the signal pattern variations caused in an indoor environment based on various use cases in the literature, along with real-life experimental studies of our own experiences related to different scenarios.

### Overview of Transfer Learning

In the IoT era, accurate positioning services are essential for LBSs that require precise location information in both outdoor and indoor environments [245,246,247,248]. IPSs are particularly important for applications such as asset tracking, smart healthcare, autonomous parking, and emergency services [246,247,248]. IPSs need to provide high accuracy, quick processing times, and low complexity, but they face challenges like signal degradation due to environmental factors, such as multipath effects and signal attenuation [249,250,251,252,253]. Fingerprint-based indoor positioning systems (FPBIPSs) are popular solutions due to their cost-effectiveness and ease of implementation, although they require extensive labeled samples for accuracy, which can be labor-intensive. Moreover, offline-calibrated fingerprint maps may not perform consistently in varying indoor conditions. To address these challenges, transfer learning (TL) methods have emerged as effective strategies, allowing models trained on one dataset to adapt to another with limited labeled data [254,255,256,257]. However, TL techniques face issues like heterogeneous feature spaces and duplicated fingerprints, which can hinder positioning accuracy. Several studies have explored various TL approaches to enhance IPS performance. For example, one study utilized a Wi-Fi-based system that adapts its localization model using real-time signal strength data, improving accuracy by 26% under environmental variations [258]. Another approach focused on using channel state information with Adaboost to mitigate performance degradation in dynamic settings [259]. Additionally, adaptive positioning systems have been developed to streamline fingerprint reconstruction and improve localization accuracy [260]. Despite advancements, challenges remain in TL applications, such as collecting sufficient training data and dealing with variations in feature spaces. Recent research has proposed algorithms that refine the source domain based on the target domain to avoid negative knowledge transfer, enhancing IPS performance [65,243,260,261]. TL methods have also been applied in fields like robot navigation and natural language processing to improve learning and adaptability [262,263,264,265,266,267,268].

In summary, the main challenges for FPBIPS include the following: (a) the inherent variability in fingerprint patterns due to environmental factors; (b) the potential duplication of fingerprints from multiple sources, leading to interference; and (c) the need for numerous training instances for accurate positioning. To overcome these, researchers propose integrating dimensionality reduction techniques, like Principal Component Analysis (PCA), with TL methods to create efficient feature spaces that enhance positive knowledge transfer. Generally, there are three research issues in applying transfer learning methods that need attention: (a) what to transfer (transferability); (b) how to transfer (learning algorithm); and (c) when to transfer (optimizing positive knowledge transfer). Moreover, the transfer learning approaches are categorized into four classes based on the “what to transfer” context: (a) instance-based, (b) feature-based, (c) parameter-based, and (d) relational-based approaches. Various transfer learning techniques have been applied for many real-world applications, including, but not limited to indoor positioning systems, robot navigation, visual categorization applications (such as object recognition, image classification, and human action recognition), modeling text classification, image classification problems, the health industry, for the prediction of health outcomes, and natural language processing.

## 6. System Architecture, Challenges, and Prospective Measures

In this section, we present an overview of the architectural considerations, challenges, and prospective measures related to the deployment of indoor positioning systems as a potential outlook for future work.

### 6.1. System Architecture

The architectural considerations for indoor positioning systems are mainly about addressing the requirements for the practicality and implementation of the system’s functionalities. Availability, scalability, security and privacy, affordability, and energy efficiency as the main architectural specifications that need to be considered.

Availability

When developing an IPS, the accessibility of the necessary infrastructure is crucial. Recent advancements in wireless communication, especially the global rollout of both 5G and 6G networks, have transformed modern lifestyles by connecting devices through the IoT [1]. Accurate location information is essential for IoT functionalities, particularly indoors, where traditional technologies like GNSS are ineffective due to signal issues. Given that people spend 70–90% of their time indoors [4], IPSs are vital for various IoT applications, including indoor rescue, precision marketing, asset tracking, mobile health services, virtual reality, and location-based social media [5,6,7,8]. Despite the challenges posed by indoor environments, IPS is a core technology for the IoT. Furthermore, the Federal Communications Commission’s incentive auction in December 2018 aimed to facilitate the transition of existing licenses to support 5G and IoT services in the upper GHz bands [269]. This highlights the growing importance of IPSs technologies like GPS, which is already critical for applications in timing, banking, and more. For IPSs to be widely adopted, the technology must be user-friendly and readily available on devices, minimizing the need for costly additional infrastructure, such as clock synchronization or specialized hardware. WLAN-based tracking systems are favorable for their universal availability and ease of deployment, making them essential for IoT applications.

2.Scalability

Scalability refers to a system’s ability to adjust positioning performance and costs based on changing application and processing demands. The goal for indoor positioning services is to replicate the success of outdoor positioning systems for navigation and tracking. To achieve large-scale indoor positioning, the technology must be affordable, energy-efficient, easy to use, and require minimal infrastructure investment. Scalability also depends on how data are collected and processed. Two main topologies exist:(1)User Device Localization: This approach processes location data on the user’s device, minimizing the impact on other users or anchor nodes. It allows the simultaneous location of many users over a large area, enhancing scalability.(2)ANBP: In this model, localization occurs on a server linked to anchor nodes, which handles multiple user requests concurrently. This can create a processing burden and may limit scalability. In summary, the scalability of an IPS is influenced by various factors, including architecture, application needs, technology, cost, and ease of use. Optimizing these elements is essential for effective implementation.

3.Security and Privacy

With advancements in communication technology, handheld devices like smartphones, laptops, and tablets have become essential for various services, including email, social networking, streaming, online meetings, education, location tracking, and health monitoring. However, security and privacy concerns for users remain significant challenges for the future of IPSs and their scalable deployment. These devices can expose user behavior and activities through constant connectivity via technologies like Wi-Fi and 4G LTE. Without robust security and privacy measures, user information may be compromised. This is particularly critical for IPS-based IoT applications, which aim to connect diverse entities in various sectors, including smart industries, healthcare, transportation, smart homes, and smart cities. To address these concerns, IPS-based IoT systems must incorporate key security features—such as confidentiality, integrity, and availability—along with privacy, authentication, and authorization measures to protect user information effectively.

(a)Authentication

Authentication is the process of verifying the identity of a user or device to confirm that it is recognized by the system. It ensures that data transmitted over networks are legitimate and identifies the entities requesting those data. There are three main methods for verifying identity: (a) knowledge-based, using known information (e.g., passwords), (b) ownership-based, using physical objects (e.g., smart cards), and (c) biometric, using unique characteristics (e.g., fingerprints) [270]. To enhance security beyond single-factor authentication, encryption-based handshake responses can be employed [270,271]. These methods fall into two categories: (a) symmetric encryption, which uses the same key for both encryption and decryption, and (b) asymmetric encryption, which uses a pair of different keys [271]. The entity being authenticated can be a user, device, application, or process. The choice of authentication method should depend on the specific use case and the confidentiality of the data involved. However, ensuring that both the requesting entity and the service provider are trustworthy remains a challenge, as there is no guarantee of reliable identification or privacy protection [272].

(b)Authorization

Authorization is the process of defining access rights or privileges for system resources, such as location information and security services. It serves as a mechanism for access control, specifying who can access what within the system. Access privileges are granted to recognized entities based on their identity and assigned access levels. For example, different access levels may be established for customers, employees, and management staff, determining which resources each group can access.

(c)Privacy

Privacy is a key security principle that protects personal information from public access, ensuring that only the user has control over their data. Unlike confidentiality, which uses encryption to shield information from unauthorized users, privacy focuses on limiting user control over specific data and preventing inferences about other valuable information. Privacy issues encompass both technical and behavioral aspects [31]. The technical aspect involves designing systems so that user location data are not stored with unique identifiers and are not transmitted in plain text over the Internet. The behavioral aspect is more challenging, as it depends on users’ willingness to share their personal details.

(d)Confidentiality

Confidentiality is a critical security feature that ensures data are secure and accessible only to authorized users or devices. In indoor environments, where numerous measurement devices (such as Wi-Fi, BLE, UWB, RFID, cameras, and LEDs) are deployed, maintaining confidentiality is essential to prevent sensitive information from being exposed to neighboring devices. Additionally, users should be informed about data management practices, including the methods used and who controls the data (the system or the user), as these factors are important for upholding confidentiality [273,274].

(e)Availability

Availability is a crucial security principle that ensures resources and services are accessible to authorized users and devices whenever needed. In many applications, real-time data requests are essential; delays in delivering these data can disrupt service provision due to security threats or server overload. One significant threat to availability is the denial-of-service (DoS) attack. Therefore, maintaining availability is vital for indoor positioning systems, requiring the implementation of effective security policies and protocols [31].

(f)Integrity

Integrity is a vital security feature that guarantees the delivery of accurate information without alterations during transmission. It ensures that data reach their destination in their original form, free from both intended and unintended interference. In indoor positioning services, maintaining data integrity is crucial; if users or service providers receive incorrect location information due to network security threats, this can lead to erroneous system operations and interruptions. To uphold data integrity in indoor environments, enhanced mechanisms, such as false data filtering schemes, should be implemented [275].

(g)Encryption

Encryption is a crucial security feature that allows only authorized parties to decrypt ciphertext and access the original information. There are three key risks associated with digital information that can be mitigated through encryption: (a) when data are stored in the user’s memory, (b) when data are stored on a server or in the cloud, and (c) when data are exchanged between two parties over a communication channel. Unauthorized third parties can intercept valuable information during transmission, posing significant security vulnerabilities. The advent of cryptographic algorithms, including symmetric and asymmetric encryption, has enhanced information security. Symmetric encryption is faster and suitable for large data sets, but it is less secure than asymmetric encryption. To strengthen security, effective key management is essential. Hybrid encryption algorithms can improve transmission security by using asymmetric encryption to protect the key, while symmetric encryption is used for the actual message, ensuring the protection of sensitive information [276].

4.Affordability

To enable large-scale indoor positioning functionalities, the underlying technology must be affordable, energy-efficient, easy to maintain, and simple to use, without requiring extra investments in infrastructure or additional hardware like separate antennas or clock synchronization devices. We anticipate that IPSs will emerge as critical infrastructure, similar to GNSS. For widespread and easy implementation, the technology should be readily available on user devices and avoid costly requirements. We concur with the perspective shared in [277] that IPS can effectively penetrate the consumer market and achieve broad deployment at an affordable price.

5.Energy efficiency

For effective indoor location functionality, energy efficiency is a crucial design consideration. While wireless communication technologies such as Wi-Fi primarily focus on communication, location services should be a secondary goal that minimizes power consumption. To enhance energy efficiency, the system must use low-power technologies and efficient algorithms with low computational complexity. Additionally, reducing the receive range can help decrease power consumption, as the transmitted power diminishes over time. However, increasing latency can negatively impact energy efficiency and user satisfaction if it affects primary service goals. Location accuracy can be compromised if too few fingerprints are used to optimize energy efficiency. In ANBP, anchors receive signals from mobile users for tracking and positioning. Although this may introduce latency due to higher computational demands on the server, it generally meets user needs effectively. Generally, the energy efficiency of an IPS is a multidimensional challenge influenced by various factors, including application requirements, algorithm efficiency, signal processing technology, localization topology, and reception range.

6.Coverage

The coverage area of the underlying technology is a crucial factor for the effective deployment of an indoor tracking system. An IPS must provide sufficient range to deliver accurate location services in large, complex spaces such as industrial factories, office buildings, shopping malls, hospitals, and parking facilities. A longer range can reduce the number of required anchor nodes, making the system more cost-effective. However, as the distance between the transmitter and receiver increases, so does the risk of signal interference. This interference can be exacerbated by the multipath effects and environmental noise typical of indoor settings. Even with a large reception range, system performance can decline due to signal attenuation and NLOS issues caused by obstacles. Thus, establishing an appropriate coverage threshold is essential, depending on the specific application and environment in which the IPS will be implemented [278].

7.Positioning Performance

Positioning performance is a critical aspect of indoor positioning systems, especially in complex environments. Fingerprint-based indoor positioning systems (FPBIPS) are often favored for their cost-effectiveness and ease of implementation, but they face significant challenges, including the following: (a) variations in signal patterns due to multipath effects, shadowing, and scattering; and (b) the need to collect a large number of tagged samples to achieve desired accuracy, which can be costly and labor-intensive. As a result, offline-calibrated fingerprint radio maps may struggle to provide consistent and accurate indoor positioning across dynamic environments. To address these signal variation issues, techniques such as transfer learning, data fusion, feature engineering, ensemble learning, and crowd sourcing can enhance location performance. It is crucial to design an indoor positioning system that optimizes available resources while maintaining an acceptable margin of error. Additionally, the system’s positioning performance is influenced by the application’s requirements, necessitating efficient signal processing technologies and effective noise reduction measures [31].

8.Robustness

Robustness refers to the ability of the system to be resistant in maintaining the discrepancies of signal patterns or fingerprint fluctuations due to the complex nature of the indoor environment, which is characterized by multipath effects, NLOS, signal scattering, and loss, which can affect the performance of the system. A robust indoor positioning system shall be able to deliver accurate positioning estimate or work consistently regardless of the possible noise or signal discrepancies.

9.Latency

Location services must often provide real-time positioning without noticeable delays, depending on the application. Fast processing is essential, and using a limited number of measurement signals can enable immediate responses. However, this limitation may compromise positioning accuracy. The latency of an IPS is influenced by how data are collected, captured, and processed. The MDBP is ideal for navigation systems, in which mobile users receive signals from anchor nodes, allowing them to independently determine their locations without interference. In contrast, ANBP is preferred for user tracking, in which anchors receive signals from mobile users. In this setup, localization occurs on the server side, which can lead to server overload and increased latency due to the computational demands of handling multiple user requests simultaneously. Thus, IPS latency is a complex issue influenced by various architectural factors. Efforts should focus on optimizing response times while considering the application requirements, signal processing technology, costs, and localization topology.

10.Reliability

Reliability is another important metric for a positioning system, which is related to robustness but is not the same. Specifically, the designation of a system as reliable means the probability that the location or position of a user or mobile device being predicted will be true with a certain acceptable threshold. In general, a higher value or higher percentage of reliability means that the true value is predicted correctly.

### 6.2. Challenges

The rise of the IoT and the widespread use of wireless communication technologies have increased interest in indoor positioning within wireless networks. Accurate location-based services are essential for various IoT applications, including smart industries, healthcare, cities, personal tracking, navigation, shopping malls, smart homes, car parking, inventory control, and location-specific routing. However, achieving accurate positioning in indoor environments presents significant challenges due to factors such as the radio environment, multipath effects, random noise, device calibration, handovers, and network technology limitations. These challenges can hinder the characterization of signals, thus restricting the effectiveness of IPSs. This section outlines the obstacles that impede the widespread adoption of IPSs and highlights the potential threats to their development as critical infrastructure, especially compared to GPS, which is widely regarded as the standard for location determination.

Multipath Effects and Noise

GPS is the standard method for location determination, known for its accuracy in outdoor environments. However, its effectiveness for indoor positioning is limited due to the complexities of indoor settings [8,9,10,11]. IPSs require higher-quality features, including greater accuracy, faster estimation times, and lower complexity for mobile devices. Among various indoor positioning technologies, FPBIPSs are the most promising solutions due to their cost-effectiveness and ease of implementation. Despite these advantages, FPBIPSs are susceptible to issues like multipath effects, shadowing, and scattering caused by indoor dynamics [12,13,14,15,16]. Additionally, signal strength attenuation due to path loss and multipath effects significantly impact location performance [17], as illustrated in Figure 8, below.

2.Radio Environment

IPSs locate users or devices in complex environments using various signal features like RSS, CSI, AOA, and TOA, especially where GPS signals are weak or blocked. GPS signals can be diminished or obstructed indoors due to factors such as (i) NLOS conditions, (ii) heterogeneous environments, (iii) multipath effects, and (iv) complex building materials that cause signal attenuation and path loss. IPSs demand higher-quality characteristics than outdoor systems, including greater accuracy, shorter estimation times, and reduced mobile device complexity. However, fluctuations in signal patterns in dynamic indoor environments negatively impact positioning performance, as illustrated in Figure 9 [243,260]. To achieve shorter processing times, fewer fingerprints may be used, which can compromise accuracy. Thus, it is crucial to design an indoor positioning system that optimizes available resources while accounting for environmental conditions and maintaining an acceptable margin of error. Additionally, system performance depends on the application’s requirements, necessitating efficient signal processing and noise reduction, which can be costly [31].

3.Heterogeneity of devices

In an indoor positioning system, the heterogeneity of devices during sampling can significantly affect positioning performance, necessitating calibration to account for variations in fingerprint data caused by differences in hardware. Maintaining consistent standards across hardware devices is a practical challenge, especially during testing, when users or devices may differ. Even measurements taken by the same device can yield varying fingerprint signatures at the same reference nodes, leading to mismatches with stored values in the radio map. This inconsistency can degrade positioning performance [279]. Additionally, the dynamic nature of indoor environments further exacerbates these challenges, contributing to overall performance degradation.

4.Lack of Standardization

A significant challenge in indoor positioning is the lack of technology standardization. While various FPBIPSs are considered promising due to their cost-effectiveness and ease of implementation, the absence of standardization hinders their widespread adoption as a critical infrastructure, similarly to GPS for outdoor positioning. With the emergence of both 5G and 6G networks and the IoT, the indoor positioning technologies have become critical infrastructure, as they are essential components of IoT applications.

5.Side effect on the service of network technology

To ensure effective and robust indoor positioning functionality, it is essential to minimize factors that interfere with the primary goal of communication. Wireless technologies like Wi-Fi primarily focus on communication, making location services a secondary objective. For a widely accessible positioning system, the underlying technology should be low-power to reduce energy consumption during user tracking. Selecting efficient algorithms with lower computational complexity can help minimize the impact of tracking on network services. Additionally, keeping the reception range minimal can reduce energy loss, as transmission power decreases over time. Any interference with the system’s primary goals may lead to user dissatisfaction or perceptions of the service as unworthy.

6.Requirement for Large Sample Size

To achieve the desired accuracy in fingerprint-based positioning, a sufficient number of labeled samples must be collected, which is often costly and labor-intensive. However, offline-calibrated fingerprint radio maps frequently fail to deliver consistent indoor positioning with high localization accuracy due to the variability of indoor environmental dynamics.

### 6.3. Prospective Measures

In this section, we highlight some of the prospective measures propose to meet some of the challenges encountered in indoor positioning.

1.Data Fusion

Numerous indoor positioning (IP) techniques have been developed to address challenges such as non-line-of-sight (NLOS) propagation and multipath effects, including TOA [7], AOA [8], CSI [9], and RSS [10]. The various wireless technologies suitable for IP, including FM radio [18], ultrasound [19], light [20], and magnetic fields [21], and radio frequency technologies such as Wi-Fi [22], RFID [23], Bluetooth [24], UWB [25], and Zigbee [26], as well as pedestrian inertial sensors [27] and infrared [28], offer distinct advantages but also limitations when used independently. Relying on a single technology can lead to significant issues if it fails, making it essential to integrate multiple sensors and measurement technologies to enhance positioning performance. Recent studies [243,280,281,282,283] highlight several key contributions: one study combines inertial sensors with short-range and long-range radio interfaces, improving accuracy through a dual-step fusion process that integrates pattern matching, PDR, and radio fingerprinting techniques [263]. Another study introduces a novel framework that integrates data-driven inertial navigation with BLE using a particle filter, achieving a reduction of up to 45.37% in mean positional error through deep learning techniques [281]. Additionally, a study fusing fingerprints from Wi-Fi, UWB, and 433 MHz technologies demonstrates an 11% accuracy improvement by combining these methods in complex indoor environments [282]. An information-theory-based method effectively combines multiple likelihood functions from various Wi-Fi access points, enhancing localization performance in environments with numerous sources [283]. Finally, a knowledge-transfer-based data fusion method for CSI enhances accuracy in dynamic indoor settings like parking lots, supported by a Cramer–Rao lower bound analysis to assess error variance [243]. Collectively, as presented in Table 9 [243,280,281,282,283], these contributions emphasize the potential of technology fusion to advance indoor localization systems, especially in dynamic and complex environments.

2.Transfer Learning

Fingerprint-based indoor positioning technologies have notable challenges, but Wi-Fi is often seen as the most promising solution for indoor positioning and location-based services due to its cost-effectiveness and ease of deployment. Key challenges for widespread adoption include (i) variability in the signal patterns affected by multipath effects, shadowing, and scattering and (ii) the need to collect a sufficient number of labeled samples for accurate positioning, which is costly and labor-intensive. As a result, offline-calibrated fingerprint wireless maps may struggle to provide consistent accuracy across different indoor environments. TL methods can help to address these challenges by leveraging labeled training data from one domain to enhance model performance in another with limited data [18,19,20]. However, two main challenges with TL techniques must be addressed: (a) differences in feature space dimensions and inherent heterogeneity and (b) the duplication of fingerprints, often recorded by multiple access points (APs), which can degrade positioning performance through negative knowledge transfer. Thus, it is essential for researchers to develop methods that minimize the labor-intensive nature of offline calibration while reducing noise from duplicated fingerprints.

Recent studies have shown significant advancements in IPSs utilizing various TL techniques [260,262,284,285,286,287,288]. One study [284] introduces a self-calibrating crowd sourcing system that employs multi-kernel deep transfer learning to continuously update fingerprint data, achieving approximately one-meter accuracy with only 50% of the updates needed by conventional methods, all while maintaining low computational complexity. Another study [285] tackles indoor localization under limited labeled data by employing a joint semi-supervised and transfer learning technique, resulting in a 43% accuracy improvement over conventional CNN methods and comparable performance with only 40% of the required data. Additionally, an LSTM-based mechanism [286] addresses positioning errors across different devices by normalizing RSS values, enhancing accuracy by 1.5 to 2 m in diverse device environments. A novel TL algorithm [287] is also presented that enhances positioning performance across multiple datasets, reducing positioning error by over 25% compared to k-NN methods. Furthermore, one study [288] frames the Wi-Fi indoor localization problem as a TL challenge, identifying key scenarios for knowledge transfer and providing a publicly available dataset for further research. Another work [260] combines heterogeneous transfer learning methods with hybrid feature selection to minimize training efforts and noise from duplicate fingerprints, achieving the lowest mean absolute error. Lastly, an online TL algorithm [262] is developed for RSS fingerprinting that integrates knowledge from both source and target domains, refining the source domain to prevent negative transfer and demonstrating robustness to environmental changes through extensive testing. Collectively, these studies highlight the potential of transfer learning and advanced algorithms in improving indoor localization accuracy and efficiency across various scenarios.

3.Feature Engineering techniques

This section discusses feature engineering techniques, such as PCA, Functional Discriminant Analysis (FDA), and correlation analysis, to address challenges related to duplicate fingerprints in complex indoor environments. For instance, the distribution of measurements at each RP can vary, leading to unbalanced datasets, as shown in Table 10 [243]. To mitigate the impact of dominant features within clusters, feature scaling techniques should be applied. Without these techniques, larger occurrence labels can overshadow smaller ones, negatively affecting modeling performance [243]. Table 10 illustrates that, despite having the same number of RPs in October 2020, the datasets were unbalanced due to uneven measurement distributions. For large datasets with high feature dimensions, using PCA is advisable to alleviate the curse of dimensionality. It is essential to standardize the features before applying PCA.

Wi-Fi fingerprint-based indoor localization methods are effective in static environments but face challenges in dynamic scenarios due to fluctuating patterns [289]. This study analyzes signal strength variations over 25 months to improve adaptive long-term Wi-Fi localization, utilizing techniques like mean-based feature selection, PCA, and FDA. The proposed algorithm, Ada-LT IP, integrates data reduction and transfer learning, enhancing accuracy while addressing multicollinearity and reducing computational complexity [289]. Another study [290] applies PCA to optimize feature selection from CSI, generating a CSI fingerprint that reflects human presence and improving classification performance in device-free settings. Additionally, PCA is used for feature selection in device-free localization, further enhancing accuracy [290]. Combining kNN and Gradient Boosting yields high prediction accuracy using the UJIndoorLoc database [291]. A novel indoor localization algorithm based on RSS improves performance through enhanced access point selection and Kernel PCA, while Robust PCA (RPCA) effectively mitigates outliers and noise [292].

(a)Principal Component Analysis

The main challenges to the widespread adoption of fingerprint-based indoor positioning can be categorized into three areas: (a) Heterogeneity in measurement distribution: Variability in fingerprint values arises from factors such as hardware differences, timing discrepancies, user and antenna orientation, and multipath conditions like diffraction, shading, fading, and interference. (b) Duplicate fingerprints: Fingerprints at grid points may be duplicated from multiple Wi-Fi access points or sensors, leading to interference between matching patterns and actual user fingerprints. This can result in irrelevant data stored in the database, degrading performance. (c) Sample size requirements: Establishing a stable relationship between signal signatures and locations requires significant instances, making large sample sizes costly. To address these challenges, we propose using feature engineering techniques to create new feature spaces that extract relevant features and facilitate positive knowledge transfer for position estimation. The new feature space should also enhance computational efficiency and cost-effectiveness, as described in [260]. PCA can reduce high-dimensional feature spaces to low-dimensional ones by selecting significant predictors while discarding less relevant information [260]. Although reducing dimensions may trade off some accuracy, the most relevant features can represent principal components that account for the majority of variation, aiding model predictions from multiple access points, as shown in Table 11 [260].

(b)Analysis of Multicollinearity Problem

The measurement independence assumption is crucial for effective sampling and robust system performance. To address multicollinearity, we employ a correlation analysis to assess whether Wi-Fi signals from multiple access points or measurements from different sensors are independent. A fingerprint value collected at a grid point may be duplicated by various Wi-Fi access points or sensors, leading to interference in fingerprint matching patterns and the generation of irrelevant, multicollinear features (e.g., X4, X6, and X6) stored in the database, which can degrade positioning performance [260]. Feature correlation analysis helps to identify multicollinearity issues. If the overall feature distribution has similar fingerprint values (e.g., −90 or −100), this indicates a lack of signal, resulting in high correlation values that do not contribute to effective modeling and may disorder the model-building process, as shown in Table 12 [260]. Measurements from multiple access points must be independent; otherwise, using several access points that lack unique characteristics will degrade positioning accuracy and increase costs.

(c)Hybrid Feature Selection

In practical scenarios, researchers may want to analyze the impact of the number of principal components or feature spaces needed for effective positioning and test the independence among those feature spaces. A hybrid feature selection approach is essential to combine different selection mechanisms for improved positioning performance. Specifically, the Wi-Fi-received signal strengths at a grid point from multiple access points should be independent. If they are highly correlated, this can degrade positioning accuracy and increase costs, as additional access points may not provide unique features for modeling variance. As shown in Table 13 [260], a hybrid feature selection approach utilizing PCA and correlation analysis can enhance positioning performance.

4.Ensemble Learning

Wi-Fi RSS-based fingerprint indoor positioning is considered the most promising solution for indoor positioning and location-based services due to the widespread availability of Wi-Fi infrastructure and the ease of extracting RSS values. However, fluctuations in signal patterns in dynamic indoor environments can negatively impact positioning performance, making it less robust. Compared to outdoor systems, indoor positioning requires high-quality attributes, including greater accuracy, shorter processing times, and reduced mobile device complexity [293,294,295]. Various measurements and algorithms have been explored in the literature to enhance positioning performance. Combining different signal metrics and algorithms can improve indoor positioning systems. For example, a hybrid fingerprint map that integrates Wi-Fi RSS values with magnetic field sensor values has shown improved classification accuracy. Experimental results indicate that ensemble approaches, which combine multiple sensor measurements and classifiers, outperform individual algorithms. This ensemble method is commonly used in machine learning to enhance estimator accuracy and has been successfully applied in indoor positioning [296,297].

5.Crowdsourcing

FPBIP involves two key phases for constructing the radio map: the training phase and the test phase. During the training phase, an offline radio map is created using fingerprints associated with known physical locations. Each fingerprint consists of a vector of RSS values from Wi-Fi access points, labeled with their true positions. Mobile users collect these RSS values and send them to the server, which employs pattern matching algorithms to compare the measurements with the fingerprints in the database to determine the user’s location. In the test phase, online positioning occurs when matching collected fingerprints against the radio map. However, this method has been criticized for being labor-intensive, costly, and susceptible to changes in the dynamic environment, which can render the radio map obsolete. Variations in fingerprint values can result from factors such as device heterogeneity, timing differences, user and antenna orientation, and multipath effects like diffraction, shading, and interference. These variations can hinder accurate positioning, leading to mismatches between online measurements and stored fingerprints. To address these challenges, various fingerprint matching patterns have been explored, which are categorized into deterministic [298,299,300] and stochastic methods [301,302,303]. Recent studies have focused on crowd sourcing to create and update radio maps, thereby reducing the need for extensive site surveying [304,305,306]. Researchers are developing algorithms that utilize crowd-sourced user traces; however, these trace matching algorithms often struggle due to the unstable posture and high-power consumption of smartphones [307,308,309].

## 7. Conclusions

In this paper, we investigate various measurement techniques and technological solutions for addressing complex indoor scenarios, offering a comprehensive overview of IPSs. We discuss the impact of the IoT as a significant factor driving indoor positioning, highlighting both the opportunities and the challenges of leveraging IoT infrastructure for this purpose. We introduce current positioning systems and evaluate them using multidimensional matrices that consider various architectural and design aspects, as well as established evaluation metrics. A general framework for IPSs is provided, summarizing recent advancements and challenges identified through real-world experiments. This framework aids researchers and practitioners in understanding the current state of indoor positioning and recognizing areas for future research. We explore challenges that hinder the widespread deployment of IPSs, emphasizing the risk in lacking a universally accepted standard, unlike GPS for outdoor positioning. We outline potential actions to address these challenges and present transfer learning approaches to enhance positioning performance, along with the associated limitations. Additionally, we propose feature engineering techniques to eliminate irrelevant features that could degrade system performance. We detail how data fusion and multisensory technologies can help to overcome real-world challenges in various IPS applications. Lastly, we provide an overview of ensemble learning and its framework for improving estimation and positioning accuracy in IPSs.

## Figures and Tables

**Figure 1 sensors-24-06876-f001:**
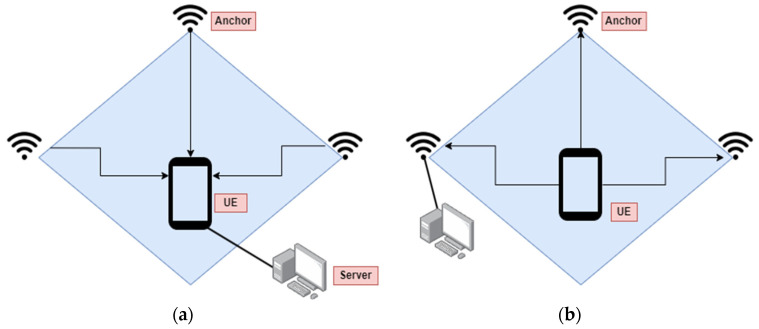
(**a**) Mobile-Device-Based IPS (MDBIP), (**b**) Anchor-Based IPS (ANBIP).

**Figure 2 sensors-24-06876-f002:**
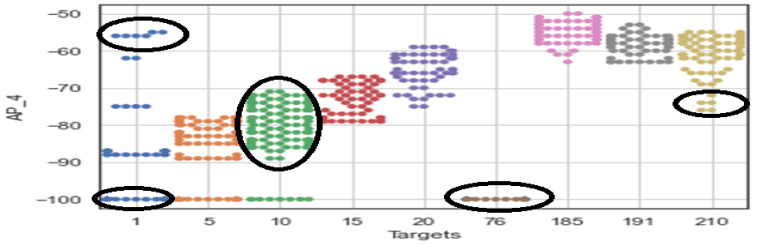
Distributions of RSS for specific access point 4 with its corresponding target values, Dataset A [65].

**Figure 3 sensors-24-06876-f003:**
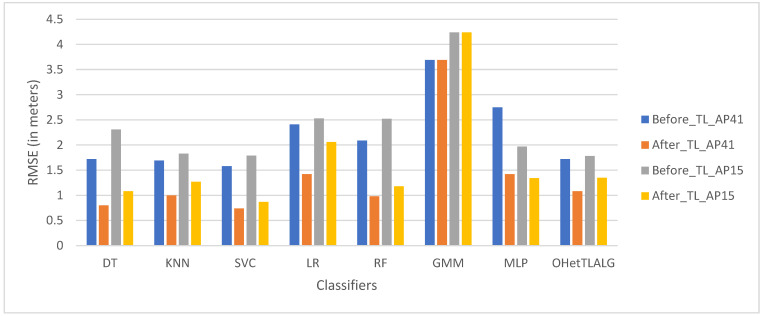
Effect of different feature spaces used to train classifiers for IPS, Dataset A [65].

**Figure 4 sensors-24-06876-f004:**
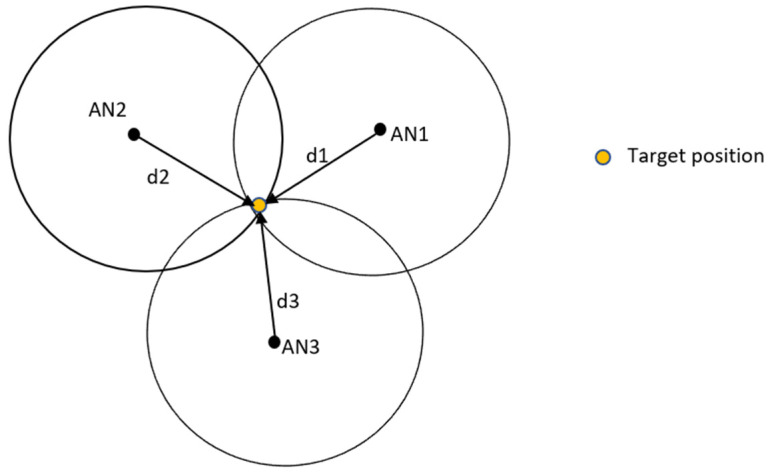
Illustration of TOA-based indoor positioning system.

**Figure 5 sensors-24-06876-f005:**
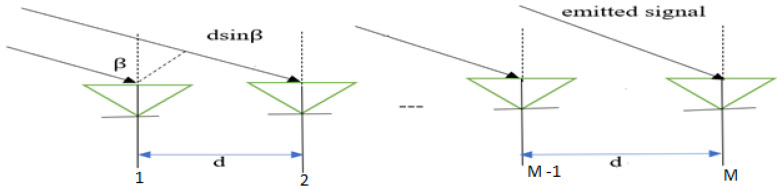
The AOA estimation scenario.

**Figure 6 sensors-24-06876-f006:**
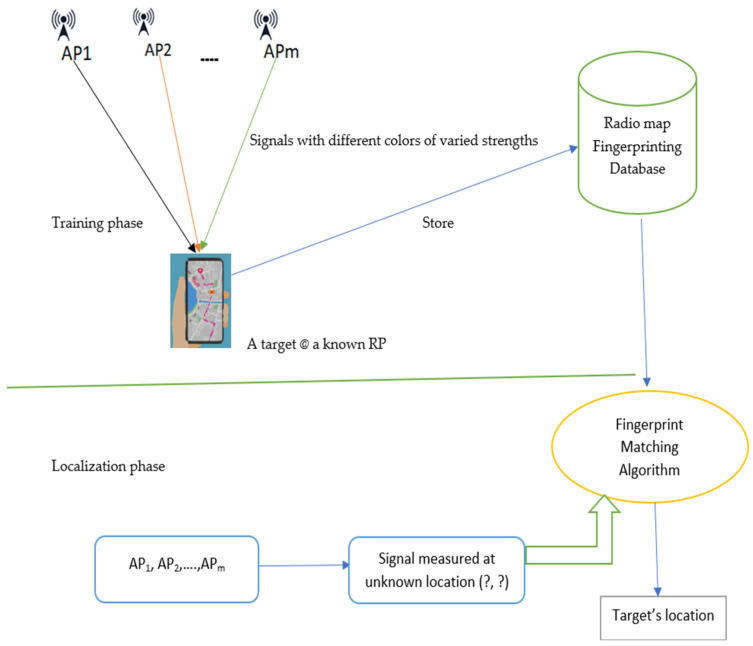
Workflow of fingerprinting-based indoor positioning system.

**Figure 7 sensors-24-06876-f007:**
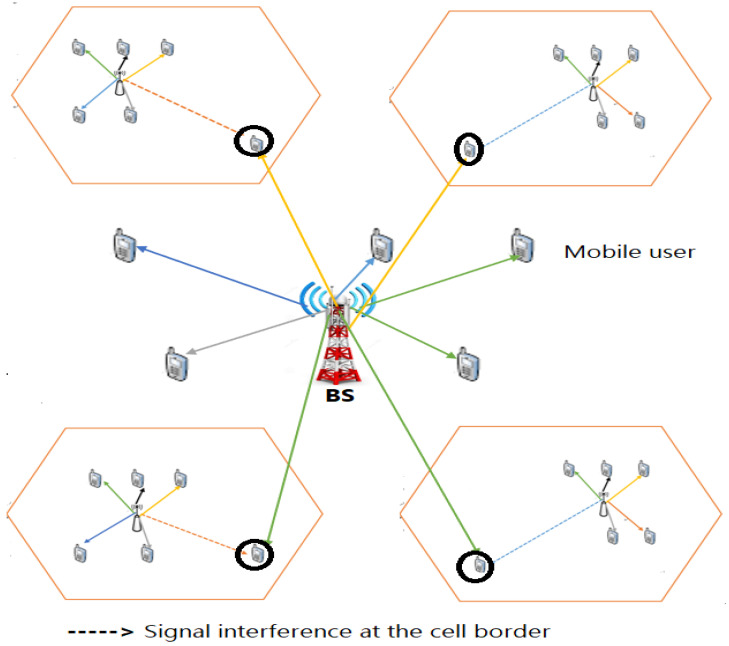
Illustration of signal interference scenarios in cellular network.

**Figure 8 sensors-24-06876-f008:**
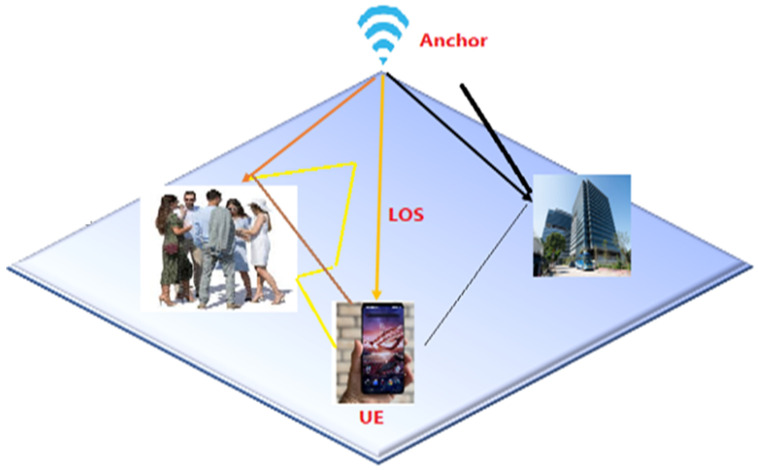
Multipath effect scenario.

**Figure 9 sensors-24-06876-f009:**
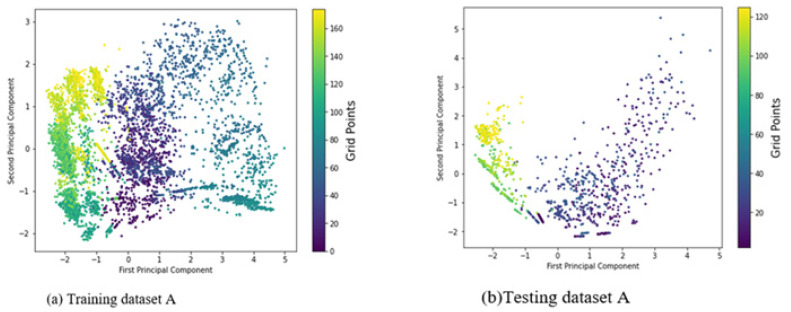
Distribution of principal components with their corresponding labels for Dataset_A [260].

**Table 1 sensors-24-06876-t001:** SWOT analysis of RSS-based fingerprinting applied for indoor positioning systems.

Strengths	Weakness	Opportunity	Threats
- Cost effectiveness	- Susceptible to environmental dynamics	- Can be extracted from almost all networks (NWs)	- Environmental dynamics is often unavoidable
- Easy implementation	- Vulnerable to real-time environmental changes - Higher interference and increased RSS fluctuation due to limited frequency range	- Easily interpreted - Scalability and improvement of technology	- Inherent heterogenous environment settings

**Table 2 sensors-24-06876-t002:** SWOT analysis of CSI-based fingerprinting applied for indoor positioning systems.

Strengths	Weakness	Opportunity	Threats
- Enhanced wireless channel metric - Relatively stable in complex indoor environment - Rich in signal information/features	- Requires extra investment - Implementation is not easy	- Significant applications demand high positioning accuracy - Need for high dimension of signal features	- Appropriate signal processing technology is required - Inherent heterogenous environment settings - Duplication of fingerprints due to signal modulation on multiple channels

**Table 3 sensors-24-06876-t003:** SWOT analysis of TOA-based fingerprinting applied for indoor positioning systems.

Strengths	Weakness	Opportunity	Threats
- Stable for real-time positioning - Lowest cost energy	- Requires extra investments or hardware for clock synchronization (1W-TOA) - Communication protocol enabling timestamps to be transmitted with the signal is needed - Delay in time to send message of calculated Target’s position to receiver (2W-TOA)	- Significant applications demands for real time positioning	- Strict clock synchronization of transceivers is required - Inherent device heterogeneity may affect the entire synchronization process - NLOS

**Table 4 sensors-24-06876-t004:** SWOT analysis of TDOA-based fingerprinting applied for indoor positioning systems.

Strengths	Weakness	Opportunity	Threats
- No extra cost of hardware for clock synchronization - Suitable for real-time positioning - Lower cost of energy	- Influence of channel noise - High computational complexity or expense - Challenge of measurement processing: nonlinearity and nonconvexity	- Involves fewer communication times - A viable method for multi-user positioning scenarios - Significant application demands for real-time positioning	- NLOS - Dynamic indoor environments - Multipath effect propagation

**Table 5 sensors-24-06876-t005:** SWOT analysis of AOA-based fingerprinting applied for indoor positioning systems.

Strengths	Weakness	Opportunity	Threats
- No time synchronization is required between transmitter and receiver - Positioning can be determined using simple geometric calculation	- Positioning performance depends on detection of transmission and reception time - Velocity of signals and angle of signal arrivals affect positioning accuracy - Accuracy is unpredictable due to NLOS, multipath effect, and random noise	- Development of multiantenna technology - Angle of orientation can be ensured using MIMO and beamforming technology	- NLOS - Random noise or dynamic indoor environments - Multipath signal arrivals

**Table 6 sensors-24-06876-t006:** SWOT analysis of PDR-based fingerprinting applied for indoor positioning systems.

Strengths	Weakness	Opportunity	Threats
- Enables real-timemotion detection and activity recognition	- Suffers from accumulated errors of inertial sensors embedded in the devices - Positioning performance degrades due to sensor error accumulation - Limited to short-range coverage	- Modern smartphones comprise numerous embedded sensors, including IMU - Smartphones collects the IMU readings, which provide basic information on user motion and inputs to the PDR	- Susceptible to speed estimation, headingdetermination, and position computation - Depends on a known initial location, user’s step length, and angle of direction at each step - Drifting effects caused by the IMU biases

**Table 7 sensors-24-06876-t007:** Comparison between model-based and fingerprinting approaches for RSS-based indoor positioning.

	Approaches	
Feature	Model-Based	Fingerprinting
Method	Uses mathematical models to estimate position.	Matches real-time RSS to pre-recorded signal data.
Accuracy	Affected by environmental factors, leading to lower accuracy.	Matches real-time RSS to pre-recorded signal data.
Setup Effort	Minimal, as no pre-survey is required.	High, due to the need for offline data collection.
Adaptability	Sensitive to changes in the environment.	Requires updates when the environment changes.

**Table 8 sensors-24-06876-t008:** Advantages and disadvantages of indoor positioning technologies.

Technology	Advantages	Disadvantages
WLAN-Based IPS	- Widely accessible with minimal infrastructure costs [63,64].	- Affected by NLOS propagation and multipath effects [61,62].
	- Uses existing Wi-Fi networks, enhancing availability [63,64].	- High costs for wireless map creation and maintenance [25,26,27].
	- Fingerprint methods outperform range-based methods in complex environments.	
BLE IPS	- Low power consumption and cost-effective for short-range applications [105].	- Limited accuracy compared to Wi-Fi [107].
	- Suitable for short-distance data transmission and IoT devices, enabling easy deployment [105,106,107,108].	- Performance degrades with obstacles and environmental factors [109].
RFID-Based IPS	- High accuracy and ability to penetrate obstacles [112].	- Depends on tag placement and density for accuracy [116].
	- Strong security features, capable of functioning in NLOS environment, and moderate costs [113,114].	- Active tags require battery maintenance [112]. - Limited range in some cases, affecting deployment [112]. - Requires infrastructure for RFID readers [112].
UWB IPS	- Centimeter-level accuracy and excellent barrier penetration [118,119].	- Higher power consumption and setup costs [121].
	- Suitable for short-distance applications [120].	- Performance affected by multipath fading and NLOS conditions [122].
INS	- Autonomous positioning without external infrastructure [124].	- Measurement errors accumulate over time (drift) [125,126].
	- Useful in environments where GPS is unreliable [127].	- Requires integration with other technologies for improved accuracy [127].
Cellular-Network-Based IPS	- Leverages existing cellular infrastructure for location estimation [129].	- Generally less accurate than dedicated indoor positioning methods [130,131].
	- Hybrid methods can enhance accuracy with GPS data [132,133].	- Signal interference and fading may hinder performance [135,136,137].
ZigBee IPS	- Energy-efficient with low data rates, suitable for medical applications [138].	- Limited range and accuracy in complex scenarios [140].
	- Cost-effective and simple to deploy [139].	- Requires regular updates to the radio map due to signal fluctuations [140].
VL-Based IPS	- High bandwidth and energy efficiency [141,142].	- Dependent on direct line-of-sight, impractical in dynamic settings [146].
	- Capable of achieving high accuracy in controlled environments [145].	- High computational demands for image processing [148].
Geomagnetic IPS	- Utilizes existing magnetic fields, no additional infrastructure needed [160,161,162,163].	- Susceptible to distortion from nearby electronic devices [166,167,168].
	- Offers temporal stability and low deployment costs [164,165].	- Calibration challenges due to sensor-reading variations [170].
Ultrasonic IPS	- High-precision positioning and cost-effective [171,172,173].	- Performance affected by noise and multipath effects [189].
	- Capable of tracking multiple users simultaneously [180].	- Requires careful deployment to minimize interference [189].
5G N/W Positioning	- Improved accuracy (within 5–10 m) [202,203,204]. - Supports dynamic environments	- Still limited in very dense areas [203,204,205] - Requires extensive infrastructure [201,202,203,204]
6G N/W Positioning	- High precision (below 1 m) [202,203,204,205,206]. - Fast update rates (hundreds of ms) [202,203,204,205,206].	- High complexity in deployment [202,203,204,205] - Requires advanced technologies like AI and UWB [202,203,204,205,206]

**Table 9 sensors-24-06876-t009:** Comparison of multi-technology data fusion techniques for IPSs.

Fusion Technique	Technologies Integrated	Advantages	Disadvantages	Reference
Dual-Step Fusion	Inertial sensors, short-range and long-range radio	Improved accuracy through pattern matching and PDR	Complexity in implementation and processing	[280]
Data-Driven Inertial Navigation with BLE	Inertial sensors, BLE	Up to 45.37% reduction in positional error using deep learning	Requires significant data for model training	[281]
Fingerprint Fusion	Wi-Fi, UWB, 433 MHz	In total, 11% accuracy improvement in complex environments	Dependency on the quality of fingerprint database	[282]
Information-Theory-Based Fusion	Multiple Wi-Fi access points	Enhances localization performance in crowded environments	Requires sophisticated algorithms and processing power	[283]
Data fusion Knowledge Transfer for CSI	Multiple Wi-Fi access points based CSI	Improves accuracy in dynamic settings, like parking lots	Involves complex data handling; requires calibration	[243]

**Table 10 sensors-24-06876-t010:** Distributions of channel state information measurements per label.

September 2020	Labels (#RPs = 225)	0	1	127	159	187	224
		0	1	2	28	29	30
Number of CSI values per Label	Dataset 1 (#Labels = 31)	793	811	742	836	806	1183
		31	32	33	34	35	36
	Dataset 2 (#Labels = 6)	899	673	820	831	842	1017
		37	38	39	59	60	61
	Dataset 3 (#Labels = 25)	931	805	803	810	1060	1254
		62	63	64	96	97	98
	Dataset 4 (#Labels = 37)	798	841	863	930	795	806
		99	100	101	132	133	134
	Dataset 5 (#Labels = 36)	822	785	797	791	804	841
		135	136	137	159	160	161
	Dataset 6 (#Labels = 27)	870	861	994	822	799	833
		162	163	164	190	191	192
	Dataset 7 (#Labels = 31)	1009	804	801	857	792	832
		193	194	195	222	223	224
	Dataset 8 (#Labels = 32)	814	804	819	799	857	832
October 2020	Labels (#RPs = 110)	0	1	127	99	107	109
		0	1	2	19	20	21
Number of CSI values per Label	Dataset 1 (#Labels = 22)	928	805	875	847	1128	903
		22	23	24	41	42	43
	Dataset 2 (#Labels = 22)	824	823	816	803	819	865
		44	45	46	63	64	65
	Dataset 3 (#Labels = 22)	815	798	808	848	829	814
		66	67	68	85	86	87
	Dataset 4 (#Labels = 22)	834	828	809	828	821	912
		88	89	90	107	108	109
	Dataset 5 (#Labels = 22)	1025	891	843	810	941	1008

**Table 11 sensors-24-06876-t011:** Effect of each PCA to the variance account of the predictive model for Dataset A.

PCAs Account for 95% Model’s Variations	Explained Variance Ratio (EVR) of Each Principal Component	
List of Principal Components	Training Data	Testing Data
PC1	42.71%	35.75%
PC2	13.89%	19.60%
PC3	12.49%	10.64%
PC4	8.863%	7.443%
PC5	7.669%	7.077%
PC6	4.756%	5.881%
PC7	4.671%	5.456%
PC8	-	4.384%

**Table 12 sensors-24-06876-t012:** Correlation analysis of the features to demonstrate the effect of multicollinearity.

	X1	X2	X3	X4	X5	X6	X7	X8	X9
X1	1.000								
X2	0.4370	1.000							
X3	0.0701	−0.0439	1.000						
X4	−0.3447	−0.4317	0.0748	1.000					
X5	0.4292	0.2644	0.2307	−0.0714	1.000				
X6	0.4376	0.3687	0.0986	−0.0794	0.3869	1.000			
X7	−0.1143	−0.1789	0.1685	0.4785	0.0297	0.0457	1.000		
X8	0.5413	0.5607	0.1154	−0.5414	0.3588	0.3585	−0.2629	1.000	
X9	−0.2844	−0.3194	0.2860	0.4452	−0.0601	−0.0791	0.3008	−0.3028	1.000

**Table 13 sensors-24-06876-t013:** Model target positions using the hybrid-based feature selection approach.

Mean Absolute Error (MAE in Meter)		
Classifiers	Before TL	After TL
Decision Tree	2.29	1.07
K-Neighbor (KNN)	1.81	1.26
Support Vector Machine (SVC)	1.75	0.98
Logistic Regression (LR)	2.51	2.04
Random Forest	2.48	1.21
Neural Network (MLP)	1.93	1.36
Proposed algorithm (Hybrid-based)	1.76	1.30

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
