# Peer review of "Theories and Methods for Indoor Positioning Systems: A Comparative Analysis, Challenges, and Prospective Measures"

_sensors, 2024, doi:10.3390/s24216876_

Round 1
Reviewer 1 Report
Comments and Suggestions for Authors
1. There is too much popularized elementary knowledge in this lengthy text. For researchers who possess rudimentary information, the fundamentals hold minimal significance, and identifying the main idea is effortless. The review paper should concentrate on the newest developments of theory and technology development.
2. The description of the localized observations in Section 2.2 is very elaborate. It is advised to properly compress it and concentrate on emphasizing the comparative comparison. Figure 4 is distorted.
3. It is advised to include a section on the indoor positioning principle algorithm, which can be broken down into sections on dead reckoning positioning, fingerprinting and other pattern matching positioning, and spatial geometric relationship positioning, between sections 2.2 Observations and 2.3 Positioning System. This would allow the PDR in section 2.2 to be moved from Positioning Observations to Positioning Principles.
4. It is advised to add two components to the positioning system in section 2.3: SLAM and 5G/6G network positioning. The placement of 5G/6G networks differs significantly from that of traditional cellular networks in terms of both mechanism and the presence of specific positioning signals. Additionally, visible light location is not the same as SLAM.
5. Rather of being explicitly quoted, the authors should redrew some straightforward schematics, such as Figure 12.
6. Reputable open-source datasets are another crucial source of research help. It is advised to include information on open source indoor positioning system datasets.
7. One crucial metric that impacts the application is positioning accuracy. Comparing and evaluating the position accuracy of various approaches and strategies is advised.
Author Response
Reviewer 1
Comments and Suggestions for Authors
Comment #1: There is too much popularized elementary knowledge in this lengthy text. For researchers who possess rudimentary information, the fundamentals hold minimal significance, and identifying the main idea is effortless. The review paper should concentrate on the newest developments of theory and technology development.
Response #1: Thank you for your insightful comment. We acknowledge your observation and we have revised the manuscript to focus more on recent theoretical advancements and cutting-edge technological developments in the field of indoor positioning systems. We have condensed the paper’s sections and removed redundant explanations of fundamental concepts, assuming the reader’s familiarity with basic principles. Instead, we have expanded on the latest research trends, challenges, and future directions in this domain to better align with the expectations of a research-oriented audience. All concerns are well taken and we believe these modifications enhance the manuscript's depth and relevance to the scientific community.
Comment #2: The description of the localized observations in Section 2.2 is very elaborate. It is advised to properly compress it and concentrate on emphasizing the comparative comparison. Figure 4 is distorted.
Response #2: Thank you for your valuable feedback. We have revised Section 2.2 by condensing the description of localized observations and placing more emphasis on the comparative analysis as suggested. Additionally, Figure 4 has been corrected to ensure proper formatting and clarity. We appreciate your constructive comments and believe these adjustments have improved the quality of the manuscript.
Comment #3. It is advised to include a section on the indoor positioning principle algorithm, which can be broken down into sections on dead reckoning positioning, fingerprinting and other pattern matching positioning, and spatial geometric relationship positioning, between sections 2.2 Observations and 2.3 Positioning System. This would allow the PDR in section 2.2 to be moved from Positioning Observations to Positioning Principles.
Response #3: Thank you for your thoughtful suggestion. In response, we have added a new section on indoor positioning principle algorithms between Sections 2.2 and 2.3, as recommended. This section covers dead reckoning positioning, fingerprinting, other pattern-matching positioning techniques, and spatial geometric relationship positioning. Additionally, the Pedestrian Dead Reckoning (PDR) previously in Section 2.2 has been relocated to this new section on Positioning Principles. We believe this restructuring enhances the clarity and logical flow of the manuscript.
Comment #4. It is advised to add two components to the positioning system in section 2.3: SLAM and 5G/6G network positioning. The placement of 5G/6G networks differs significantly from that of traditional cellular networks in terms of both mechanism and the presence of specific positioning signals. Additionally, visible light location is not the same as SLAM.
Response #4: Thank you for your suggestion. We have added the components of SLAM and 5G/6G network positioning to Section 2.3. We've also clarified the significant differences between the positioning mechanisms of 5G/6G networks compared to traditional cellular networks, as well as the distinctions between visible light positioning and SLAM. We appreciate your feedback in enhancing the clarity and depth of our discussion.
Comment #5. Rather of being explicitly quoted, the authors should redrew some straightforward schematics, such as Figure 12.
Response #5: Thank you for your suggestion. We have added tables and figures to describe and summarize the text.
Comment #5. Reputable open-source datasets are another crucial source of research help. It is advised to include information on open-source indoor positioning system datasets.
Response #5: Thank you for your insightful comment. In our future work, we will include detailed information on reputable open-source datasets for IPSs. We recognize the importance of these datasets for research and development, as they provide valuable resources for testing algorithms and validating positioning methods. We will compile a list of notable open-source datasets, along with descriptions of their features, accessibility, and potential applications in a separate work.
Comment #6. One crucial metric that impacts the application is positioning accuracy. Comparing and evaluating the position accuracy of various approaches and strategies is advised.
Response #6: Thank you for comment and we have made comparison based on the suggested metric.
Reviewer 2 Report
Comments and Suggestions for Authors
This paper widely reviews different theories and methods for indoor position systems from different perspectives. Despite the great amount of information presented, there is poor paper structuring and a very limited number of insights for directing future research. Below are some recommendations and/or comments from the reviewer if the authors would like to make some improvements on the current submission.
Major comments:
1. This paper provides readers with a great amount of information about indoor positioning technologies from different perspectives. However, due to insufficient paper organization, the current paper is too lengthy. The authors are recommended to add more figures or tables to summarize the information collected from different reviewed literature and then reduce the texts for greater presentation and paper structuring.
2. Review papers should also follow certain methodology to ensure their validity. However, this review paper misses a section that explicitly explains what methodology that authors used to search for, screen, extract, summarize, and discuss the information from existing relevant sources. Meanwhile, what are those sources (e.g. Scopus)?
3. In section “2.3. Positioning Technologies,” the authors are recommended to add a table to summarize the advantages and disadvantages of each of those ten IPS methods/technologies.
4. In section “6.1 System Architecture,” how does each of the IPS methods/technologies mentioned in section 2.3 perform against the different functionalities? The authors can add one table to answer this question.
5. In “6.2 Challenges,” what are the proposed solutions to each of the challenges?
6. How would this review paper direct future studies? The authors can add one section before the conclusion section to highlight the future research opportunities.
Minor comments:
1. Line 64 and 94: RFID and CSI are used before they are defined
Comments on the Quality of English LanguageOverall, it is fine except some minor editing being required.
Author Response
Reviewer 2
Comments and Suggestions for Authors
This paper widely reviews different theories and methods for indoor position systems from different perspectives. Despite the great amount of information presented, there is poor paper structuring and a very limited number of insights for directing future research. Below are some recommendations and/or comments from the reviewer if the authors would like to make some improvements on the current submission.
Major comments:
Comment 1. This paper provides readers with a great amount of information about indoor positioning technologies from different perspectives. However, due to insufficient paper organization, the current paper is too lengthy. The authors are recommended to add more figures or tables to summarize the information collected from different reviewed literature and then reduce the texts for greater presentation and paper structuring.
Response #1: Thank you for your comment and in the revised manuscript we have addressed all issues as per your suggestion.
Comment 2. Review papers should also follow certain methodology to ensure their validity. However, this review paper misses a section that explicitly explains what methodology that authors used to search for, screen, extract, summarize, and discuss the information from existing relevant sources. Meanwhile, what are those sources (e.g., Scopus)?
Response #2: Thank you for your insightful comment and all the references we cited are main from IEEE, Sensors, ACM, Wiley, and all of them are SCI indexed.
Comment 3. In section “2.3. Positioning Technologies,” the authors are recommended to add a table to summarize the advantages and disadvantages of each of those ten IPS methods/technologies.
Response #3: Thank you for your comment and in the revised manuscript we have added tables and Figures to summarize the text and specifically we summarize the advantages and disadvantages of each of those ten IPS methods/technologies using Table.
Comment 4. In section “6.1 System Architecture,” how does each of the IPS methods/technologies mentioned in section 2.3 perform against the different functionalities? The authors can add one table to answer this question.
Response #4: Thank you for your insightful comment and in the revised manuscript we addressed this comment using comparative analysis in Tables.
Comment 5. In “6.2 Challenges,” what are the proposed solutions to each of the challenges?
Response #5: Thank you for your insightful comment and we have clearly described in section 6.3. Prospective Measures
Comment 6. How would this review paper direct future studies? The authors can add one section before the conclusion section to highlight the future research opportunities.
Response #6: Thank you for your insightful comment and we believe that section 6.3. Prospective Measures are the future research directions with different views and methods are forwarded to be explored in future.
Round 2
Reviewer 1 Report
Comments and Suggestions for Authors
Thanks for the author‘s careful revision. I am quite satisfied with this revision and recommend its acceptance.
Author Response
Dear Reviewer,
Many thanks for your positive feedback and recommendation for acceptance. We appreciate your acknowledgment of our careful revisions and are glad to know that our work meets your expectations. Your support is invaluable to us, and we look forward to the opportunity to share our findings with the broader community.
Reviewer 2 Report
Comments and Suggestions for Authors
The authors addressed most of the reviewer's comments/recommendations except the one that requires the authors to add a short section to discuss the followed methodology by this review paper.
Author Response
Dear Reviewer,
Thank you for your feedback. We appreciate your acknowledgment of our revisions and would like to clarify that we have added a dedicated portion discussing the methodology followed in our paper to the Introduction section. This addition outlines our systematic approach to searching, screening, extracting, summarizing, and discussing the relevant literature, enhancing the clarity and rigor of our work. Thank you for your guidance, and we hope the revisions meet your expectations.
Here below is the revised version added to the introduction section marked with a yellow color:
To ensure the validity and rigor of this paper, we employed a systematic methodology encompassing the searching, screening, extracting, summarizing, and discussing of information from relevant sources. A comprehensive literature search was conducted across multiple databases, primarily focusing on IEEE (IEEE Xplore Digital Library), Sensors (MDPI), ACM (ACM Digital Library), and Wiley (Wiley Online Library), all of which are SCI-indexed. These databases were selected for their extensive coverage of peer-reviewed literature in engineering, technology, and applied sciences, particularly in the field of IPSs, facilitating the identification of influential works. We established clear inclusion criteria, focusing on peer-reviewed articles published within the last ten years related to IPSs (for almost all references we cited), while excluding non-English articles and non-peer-reviewed literature. The initial search results were screened based on titles and abstracts, with full-text reviews confirming relevance to our study objectives. Systematic data extraction was performed to gather key findings, methodologies, and pertinent statistics. The extracted data were thematically organized to summarize trends and significant findings, enhancing the understanding of the existing research landscape. Finally, the findings were discussed in relation to the literature, identifying areas for future research and implications for practice. This systematic approach, supported by SCI-indexed sources, ensures the quality and relevance of the materials included in our review.